# SPOP mutation induces replication over-firing by impairing Geminin ubiquitination and triggers replication catastrophe upon ATR inhibition

Jian Ma[1,2,3,10], Qing Shi[4,10], Gaofeng Cui[3,10], Haoyue Sheng[1,2,3], Maria Victoria Botuyan [3,5], Yingke Zhou[3], Yuqian Yan[3], Yundong He[3], Liguo Wang [6], Yuzhuo Wang [7], Georges Mer [3,5,8✉], Dingwei Ye[1,2✉], Chenji Wang [4✉] & Haojie Huang [3,5,9✉]

Geminin and its binding partner Cdt1 are essential for the regulation of DNA replication. Here we show that the CULLIN3 E3 ubiquitin ligase adaptor protein SPOP binds Geminin at endogenous level and regulates DNA replication. SPOP promotes K27-linked non-degradative poly-ubiquitination of Geminin at lysine residues 100 and 127. This poly-ubiquitination of Geminin prevents DNA replication over-firing by indirectly blocking the association of Cdt1 with the MCM protein complex, an interaction required for DNA unwinding and replication. SPOP is frequently mutated in certain human cancer types and implicated in tumorigenesis. We show that cancer-associated SPOP mutations impair Geminin K27-linked poly-ubiquitination and induce replication origin over-firing and re-replication. The replication stress caused by SPOP mutations triggers replication catastrophe and cell death upon ATR inhibition. Our results reveal a tumor suppressor role of SPOP in preventing DNA replication over-firing and genome instability and suggest that SPOP-mutated tumors may be susceptible to ATR inhibitor therapy.

[1] Department of Urology, Fudan University Shanghai Cancer Center, Shanghai 200032, China. [2] Department of Oncology, Shanghai Medical College, Fudan University, Shanghai 200032, China. [3] Department of Biochemistry and Molecular Biology, Mayo Clinic College of Medicine and Science, Rochester, MN 55905, USA. [4] State Key Laboratory of Genetic Engineering, MOE Engineering Research Center of Gene Technology, School of Life Sciences, Fudan University, Shanghai 200438, China. [5] Mayo Clinic Cancer Center, Mayo Clinic College of Medicine and Science, Rochester, MN 55905, USA. [6] Divison of Computational Biology, Mayo Clinic College of Medicine and Science, Rochester, MN 55905, USA. [7] Department of Experimental Therapeutics, BC Cancer Research Centre, Vancouver, BC, Canada. [8] Department of Cancer Biology, Mayo Clinic College of Medicine and Science, Rochester, MN 55905, USA. [9] Department of Urology, Mayo Clinic College of Medicine and Science, Rochester, MN 55905, USA. [10] These authors contributed equally: Jian Ma, Qing Shi, Gaofeng Cui. ✉email: mer.georges@mayo.edu; Dwyeli@163.com; chenjiwang@fudan.edu.cn; huang.haojie@mayo.edu

Genomic stability relies on precise genome replication. Tens of thousands of DNA replication start sites must be established during each cell cycle to ensure the accurate and complete duplication of more than 3 billion base pairs of DNA in the human genome[1]. To ensure accurate progress in DNA replication, licensing of this process is initiated by assembly of the pre-replicative complex (pre-RC) on replication origins at G1 phase. After the G1/S transition, origins are not to be re-licensed or reactivated for the remainder of the cell cycle. In yeast, origin reactivation is a driver of gene amplification, copy number variation, and aberrant chromosome segregation[2,3]. In mammalian cells, it causes chromosomal breaks and activation of the DNA damage response[4,5].

The pre-RC is composed of the origin recognition complex (ORC) including Cdc6, Cdt1, and the mini-chromosome maintenance (MCM) proteins[6]. ORC binds origins of replication and recruits Cdc6 at the M/G1 transition. Cdc6-bound ORC recruits Mcm2-7 in complex with Cdt1 at the origins of DNA replication[7–9]. Once the pre-RC is assembled, origins are licensed for replication in S phase and are ready to be fired. Cdt1 activity is limited to G1 through the control of its synthesis, degradation, and activity. The low level of Cdt1 in the early S phase is thought to result from targeted degradation[10–12] whereas its higher level in G2 phase is caused by its stabilization[13]. However, the increase of Cdt1 in late S and early G2 poses a potential risk of replication origin over-firing and re-replication, which could occur if there were residual activity of the DNA-replicating enzymes in G2. The activity of Cdt1 is tightly controlled by Geminin[14], a re-replication inhibitory factor, which directly binds to and suppresses the replication-stimulating function of Cdt1[15]. Geminin is an unstable protein that is targeted for degradation by the anaphase-promoting complex (APC)[16]. Both Geminin and Cdt1 are expressed at high levels in late S and G2 phases, where Geminin binds Cdt1 and prevents DNA re-replication[15,17,18]. Furthermore, Geminin controls the basal level of Cdt1 and induces its accumulation during mitosis by inhibiting its ubiquitin-dependent proteolysis[19]. Thus, it is proposed that Geminin has both negative and positive roles in pre-RC formation, indicating that the protein level of Geminin may not be the sole key regulating mechanism in controlling its function to ensure proper DNA replication.

Two Geminin molecules self-associate via a coiled-coil domain to form a homodimer[20–22], or possibly a tetramer as suggested by crosslinking experiments[23]. The Geminin dimer forms a heterotrimer with Cdt1, which is required for the inhibition of Cdt1 function[20–22]. However, the Geminin/Cdt1 complex may exist in different states since Geminin/Cdt1 can be replication-active or -inactive depending on the stage of the cell cycle[24,25]. A Geminin/Cdt1 complex heterotrimer-heterohexamer transition model was proposed to explain the active and inactive states of Cdt1[26]. Even if Cdt1/Geminin has been extensively studied, there is no conclusive understanding of how this complex is regulated.

The SPOP gene encodes a substrate-binding adaptor subunit of the CULLIN3 (CUL3)-RING box 1 (RBX1) E3 ubiquitin ligase (CRL) complex. SPOP is implicated in oncogenesis since it is frequently mutated in human cancers such as prostate and endometrial cancers[27–29]. Notably, almost all SPOP mutations detected thus far (except one mutant) in prostate cancer patients are hemizygous mutations[27,28,30]. Increasing evidence indicates that prostate cancer-derived SPOP mutants function in a dominant-negative manner[31–33], which is consistent with the findings from biochemical and structural studies showing that the SPOP protein can form a dimer or oligomer via its BTB domain and BACK domain[34,35]. Several cancer-relevant proteins have been identified as the substrates of SPOP, such as androgen receptor (AR), SRC-3, TRIM24, and BRD4, and these proteins are

aberrantly upregulated in SPOP-mutated PCa cells and patient tissues[31–33,36–40]. SPOP is also implicated in regulating genomic stability[41–43]. However, how SPOP precisely controls genomic stability remains poorly understood.

In the present study, we demonstrate that SPOP functions as an E3 ubiquitin ligase that binds to Geminin abundantly at S phase and catalyzes K27-linked non-degradative poly-ubiquitination of Geminin. We show that SPOP-dependent poly-ubiquitination of Geminin blocks MCM binding to Cdt1. This process prevents over-firing of DNA replication. Cancer-associated SPOP mutations impair DNA replication surveillance and cause replication origin over-firing and re-replication, thereby increasing replication stress and sensitizing cancer cells to ATR inhibition.

## Results

**Identification of the DNA replication factor Geminin as a SPOP-binding protein.** Both The Cancer Genome Atlas (TCGA) prostate cancer dataset[28] and the whole-genome sequencing data of an independent cohort[44] show that SPOP mutant tumors display higher genome alterations than SPOP wild-type (WT) tumors (Fig. 1a, b). While a handful of previous studies suggest that SPOP deregulation may lead to genomic instability[41–43], no study has directly examined the impact of SPOP mutations on DNA replication. Hjorth-Jensen reported that SPOP knockdown led to reduced transcription of genes important for DNA repair and replication including BRCA2, ATR, CHK1, and RAD51[43]. However, the TCGA data show that none of these genes are downregulated in SPOP-mutated prostate cancer patient samples (Supplementary Fig. 1a–d), and the mRNA levels of these genes are negatively correlated to SPOP mRNA expression in prostate cancers (Supplementary Fig. 1e-h). The patient specimen data stress that SPOP mutations may affect genome alterations via regulating DNA replication through mechanism(s) independent of transcription of replication regulatory genes.

The majority (>97%) of SPOP mutations detected in prostate cancer patient samples is localized in the substrate-binding MATH domain[28,30], suggesting that tumorigenesis linked to SPOP mutations originates from deregulation of SPOP substrate ubiquitination. To determine how many of the proteins that directly bind SPOP are DNA replication and repair factors (Supplementary Table 1), we performed cluster analysis of the yeast two-hybrid (Y2H) screen results (Supplementary Table 2) and SPOP interactome we generated through tandem affinity purification and mass spectrometry (Supplementary Table 3). We found that Geminin is the only DNA replication factor among the SPOP-interacting proteins identified by the two independent methods (Fig. 1c and Supplementary Fig. 1i), implying that Geminin is a SPOP-interacting protein. Co-immunoprecipitation (co-IP) assays confirmed that both ectopically expressed and endogenous SPOP interacted with Geminin in 293T cells and PC-3 prostate cancer cells (Fig. 1d, e and Supplementary Fig. 1j). Notably, co-IP assay showed that SPOP associated with Cdt1 to the same extent as Geminin whereas SPOP association with MCM proteins was undetectable (Fig. 1d, e). We also generated a Geminin mutant by deleting the Cdt1-binding region (amino acids 82–160)[20–22] for a co-IP assay. As expected, this deletion mutant lost the ability to bind Cdt1, but was able to bind SPOP (Supplementary Fig. 1k), suggesting that SPOP interacts with Geminin in a manner independent of Cdt1. In support of this notion, binding assay further confirmed that in vitro translated SPOP directly binds to GST-Geminin purified from E. coli bacteria (Supplementary Fig. 1l). Together, our data demonstrate that SPOP directly binds Geminin and that their interaction occurs at the endogenous level.

Proteins bound by SPOP usually harbor a SPOP-binding consensus sequence (SBC, $\Phi$-$\pi$-S-S/T-S/T where $\Phi$ is a nonpolar

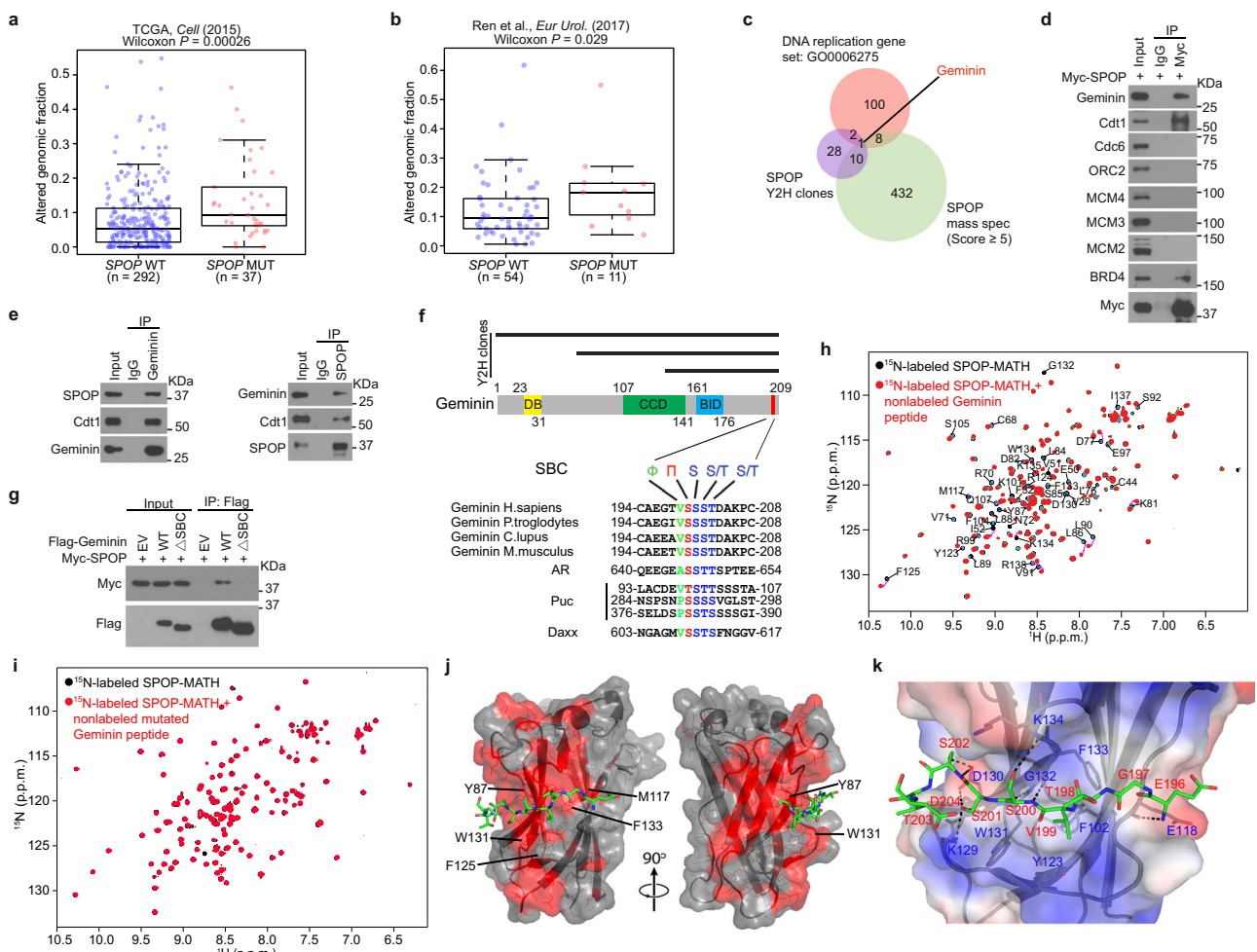

**Fig. 1 Identification of the DNA replication factor Geminin as a SPOP-binding protein. a, b** Analysis of altered genomic fraction in prostate cancer patient specimens with wild-type (WT) (blue dots) or mutant (MUT) (red dots) SPOP in The Cancer Genome Atlas (TCGA) dataset[28] (**a**) and an independent prostate cancer whole-genome sequencing dataset[44] (**b**). Data were generated using the cBioPortal (https://www.cbioportal.org/) online tool. Boxplots show median, interquartile ranges, and all data points. *P* values were calculated using one-sided Wilcoxon rank-sum test. **c** Venn diagram showing the overlap of yeast two-hybrid screen data from Fudan University (Shanghai), mass spectrometry-based SPOP interactome from Mayo Clinic, and the DNA replication gene set. **d** Co-IP analysis of DNA replication-related proteins in 293T cells transfected with Myc-SPOP. **e** Co-IP analysis of endogenous proteins in PC-3 cells using the indicated antibodies. **f** A Geminin domain structure diagram showing a putative evolutionarily conserved SBC motif (in red) located at the C-terminal end of Geminin (middle and bottom). All three Y2H clone clusters contain this motif (top). **g** Co-IP analysis of indicated proteins in 293T cells transiently transfected with Myc-SPOP-WT or ΔBTB mutant. **h, i** Perturbations in the ¹H-¹⁵N HSQC NMR spectra of ¹⁵N-labeled SPOP MATH upon titration with unlabeled WT (**h**) or mutated (**i**) Geminin peptide (amino acids 195–209). **j** Crystal structure of SPOP-MATH (surface representation) in complex with the Geminin peptide (amino acids 195–209) in stick representation. Peptide residues 196–204 were modeled in the electron density. The red surface corresponds to SPOP residues for which chemical shift perturbations were detected in the ¹H-¹⁵N HSQC NMR spectrum of ¹⁵N-labeled SPOP MATH titrated with unlabeled Geminin peptide. Selected SPOP residues frequently mutated in cancer patients are labeled. **k** Representation of Geminin peptide and SPOP residues at the binding interface. Putative intermolecular hydrogen bonds are shown in black dashes. The potential electrostatic surface of SPOP is shown in blue and red for positive and negative charges, respectively. Source data are provided in this paper or Mendeley database (10.17632/8n7xt5rkhc.1). Similar results for (**d**), (**e**), (**g**) panels were obtained in two independent experiments.

residue and π is a polar residue)[34]. The C-terminal segment ¹⁹⁹VSSST²⁰³ is the only putative SBC motif in Geminin. All three clusters of Y2H positive clones contain this motif (Fig. 1f). Deletion of these five amino acids completely abolished SPOP interaction with Geminin in 293T cells (Fig. 1g), indicating that ¹⁹⁹VSSST²⁰³ is a functional SBC motif in Geminin.

Nuclear magnetic resonance (NMR) spectroscopy confirmed this binding motif and better defined the binding interface in SPOP. The Geminin WT SBC-containing peptide (¹⁹⁵AEGTVSSSTDAKPCI²⁰⁹) and the SBC-alanine mutant counterpart (¹⁹⁵AEGTVAAAADAKPCI²⁰⁹) were tested for interaction with the recombinant SPOP MATH domain (amino acids 28–166) purified from *E. coli*. Upon addition of non-labeled Geminin WT peptides to ¹⁵N-labeled SPOP-MATH domain, there were multiple chemical shift perturbations in the ¹H-¹⁵N heteronuclear single quantum coherence (HSQC) spectrum of SPOP MATH (Fig. 1h). In contrast, there were almost no changes in the ¹H-¹⁵N HSQC spectrum of SPOP MATH upon addition of Geminin SBC-alanine mutant peptide (Fig. 1i), demonstrating that the SBC motif ¹⁹⁹VSSST²⁰³ is essential for SPOP-Geminin interaction. We also determined the X-ray crystal structure of SPOP-MATH domain in complex with the Geminin peptide ¹⁹⁵AEGTVSSSTDAKPCI²⁰⁹ to a resolution of 3.4 Å (Table 1). There are two copies of SPOP-Geminin complex in the

**Table 1 Data collection and refinement statistics for SPOP-MATH-Geminin peptide (PDB entry 7KLZ).**

| Data collection[a] | |
| --- | --- |
| Space group | P 41 21 2 |
| Cell dimensions | |
| *a, b, c* (Å) | 103.06, 103.06, 131.81 |
| *α, β, γ* (°) | 90.00, 90.00, 90.00 |
| Resolution (Å) | 47.99–3.40 (3.52–3.40)[b] |
| $R_{merge}$ | 0.096 (1.27) |
| $I/\sigma(I)$ | 15.24 (1.93) |
| Completeness (%) | 99.8 (100) |
| Redundancy | 37.0 (38.5) |
| Refinement | |
| Resolution (Å) | 47.99–3.40 |
| No. reflections | 10264 |
| $R_{work}/R_{free}$ | 0.21/0.24 |
| No. atoms | 2307 |
| Protein | 2297 |
| Ligand/ion | 10 |
| Water | 0 |
| B-factors | |
| Protein | 89.0 |
| Ligand/ion | 147.6 |
| Water | |
| R.m.s. deviations | |
| Bond lengths (Å) | 0.002 |
| Bond angles (°) | 0.48 |

[a]One crystal was used for structure determination.
[b]Values in parentheses are for highest-resolution shell.

asymmetric unit. The Geminin peptide electron density is detectable for each complex, within which we could fit nine Geminin residues from Glu196 to Asp204 (Supplementary Fig. 1m). Consistent with our NMR data, the peptide lies in a shallow groove across SPOP surface (Fig. 1j). The binding mode in SPOP-Geminin is similar to those in crystal structures of SPOP complexed with other peptides[34]. Geminin Val199 sits in a hydrophobic pocket formed by SPOP Phe102, Tyr123, Trp131, and Phe133, while Geminin Ser200, Ser201, and Ser202 probably participate in a network of intermolecular hydrogen bonds with SPOP residues Lys129, Asp130, Gly132, and Lys134 (Fig. 1k). The Geminin [199]VSSST[203] motif contributes 8 out of 9 putative intermolecular hydrogen bonds between SPOP and Geminin. Some of the SPOP residues for which there are NMR chemical shift perturbations (Fig. 1j) are frequently mutated in patients with prostate cancer (Tyr87, Phe125, Trp131, and Phe133) and endometrial cancer (Met117)[27,29].

**SPOP promotes K27-linked non-degradative poly-ubiquitination of lysine residues 100 and 127 in Geminin.** Increased expression of SPOP WT largely enhanced Geminin poly-ubiquitination in 293T cells, and this ubiquitination had no effect on Geminin protein level (Fig. 2a). Geminin ubiquitination by SPOP was confirmed by different methods (Supplementary Fig. 2a–c). SPOP could not induce poly-ubiquitination of Geminin SBC deletion mutant (Supplementary Fig. 2d). Knockout of endogenous SPOP by CRISPR-Cas9 greatly attenuated Geminin poly-ubiquitination in 293T cells (Fig. 2b and Supplementary Fig. 2e) and such effect was reversed by restored expression of SPOP (Fig. 2b), suggesting a specific effect of SPOP on Geminin poly-ubiquitination. Given that SPOP overexpression or knockdown did not affect Geminin protein level (Fig. 2a and Supplementary Fig. 2f–i), we sought to determine the ubiquitin chain linkage type(s) of SPOP-mediated poly-ubiquitination of Geminin. To this end, we expressed WT ubiquitin, single lysine residue-only or lysine-null mutants in 293T cells and showed that SPOP specifically

augmented K27-linked poly-ubiquitination of Geminin (Fig. 2c). To determine which lysine residues of Geminin are ubiquitinated by SPOP, we transfected 293T cells with Flag-tagged Geminin in combination with SPOP and ubiquitin and collected cells for mass spectrometry. Ubiquitination at lysine residues 27, 50, 100, and 127 in Geminin was detected by mass spectrometry (Fig. 2d). Mutagenesis analysis showed that only mutation of Lys100 and Lys127 substantially reduced SPOP-dependent Geminin poly-ubiquitination (Fig. 2e). Mutations of both Lys100 and Lys127 completely abolished SPOP-induced poly-ubiquitination of Geminin (Fig. 2f). These data indicate that SPOP promotes K27-linked poly-ubiquitination on Geminin Lys100 and Lys127.

**Prostate cancer-derived SPOP mutants fail to promote Geminin poly-ubiquitination.** SPOP mutations in prostate cancers mainly occur in the MATH domain, which is responsible for substrate binding[28,30,45] (Fig. 3a). Using co-IP assays, we showed that SPOP ΔMATH mutant lost the ability to bind to and promote ubiquitination of Geminin while CUL3-binding-deficient mutant ΔBTB, which cannot ubiquitinate substrates, retained the ability to bind Geminin (Fig. 3b). As expected, SPOP ΔBTB failed to promote poly-ubiquitination of Geminin (Fig. 3c). Based upon our NMR spectroscopy and X-ray crystallography results that Geminin interacts with SPOP-MATH domain on a surface frequently mutated in prostate cancer patients (Fig. 1h–k), we examined whether prostate cancer-associated mutations in SPOP would impair its ability to promote Geminin poly-ubiquitination. We generated 11 prostate cancer-associated SPOP mutants. Co-IP assays demonstrated that the Geminin binding ability of all 11 SPOP mutants was largely impaired compared with SPOP WT (Fig. 3d). SPOP-mediated poly-ubiquitination of Geminin was also markedly attenuated by these mutations (Fig. 3e). Over-expressing SPOP-WT or SPOP mutants have no influence on Geminin protein level (Supplementary Fig. 2f). Furthermore, we showed that in vitro Geminin poly-ubiquitination by SPOP was abolished in the context of SPOP F102C and F133V, two SPOP mutations most frequently found in prostate cancer patient specimens (Fig. 3f). Thus, prostate cancer-associated SPOP mutants fail to promote Geminin poly-ubiquitination.

Almost all the SPOP mutations detected so far (with one exception) are hemizygous mutations and act in a dominant-negative fashion[27,30,45]. To mimic the pathophysiological conditions in patients, we introduced the mutated allele of SPOP into prostate cancer cell lines that do not contain endogenous mutated SPOP. We transfected SPOP mutant F133V into BPH1 and DU145 cell lines and treated the cells with or without hydroxyurea (HU), a replication inhibitory agent, to see whether Geminin protein level changes in a different context. Expression of F133V did not alter the protein level of Geminin or its partner Cdt1 (Supplementary Fig. 2i). Thus, SPOP mutations impair non-degradative poly-ubiquitination of Geminin.

We also noticed that there are two Geminin mutations in lung and colon cancer patient samples of the TCGA cohort that occur in the SBC motif (S202F and T203 nonsense mutation) (https://cancer.sanger.ac.uk/cosmic). We demonstrated that the Geminin S202F mutant failed to be bound and poly-ubiquitinated by SPOP (Fig. 3g, h). These data indicate that both Geminin and SPOP could be pathophysiologically mutated and contribute to disease progression.

**SPOP-mediated poly-ubiquitination of Geminin inhibits MCM protein binding to Cdt1.** To investigate how SPOP-mediated ubiquitination influences the DNA replication regulatory function of Geminin, we examined whether Geminin ubiquitination affects its interaction with Cdt1. Co-IP data showed that transient

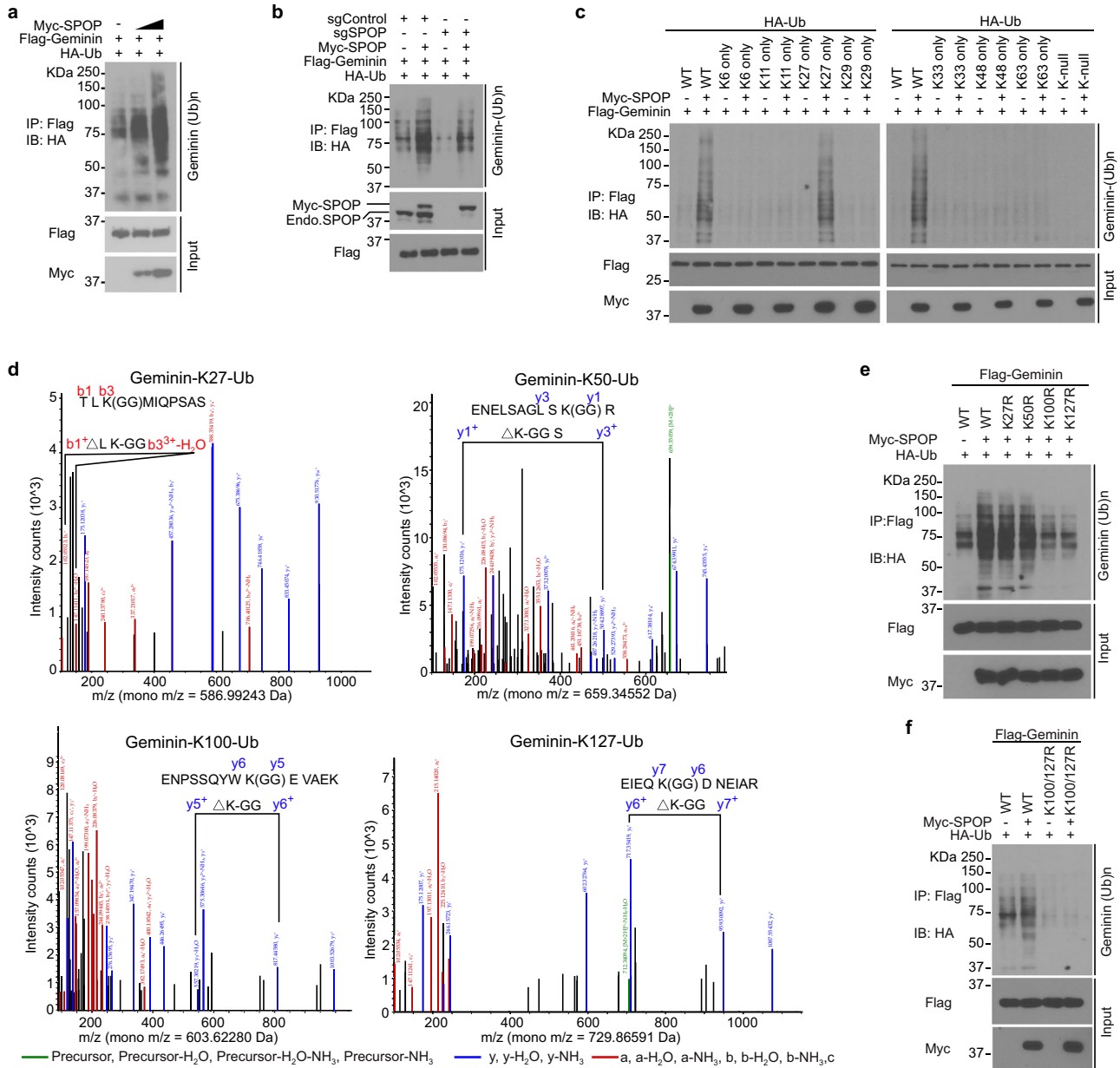

**Fig. 2 SPOP promotes ubiquitin lysine 27-linked poly-ubiquitination of lysine 100 and 127 in Geminin. a** 293T cells were transfected with increased Myc-SPOP WT in combination with Flag-Geminin and HA-Ub and harvested for IP and WB. **b** Control or SPOP knockout 293T cells were transfected with the indicated plasmids and harvested for IP and WB. **c** 293T cells transfected with plasmids for HA-tagged WT or mutated Ub in combination with other indicated constructs and harvested for IP and WB. **d** Mass spectrometry analysis reveals Geminin ubiquitination at lysine residues 27, 50, 100, and 127. **e**, **f** 293T cells were transfected with Flag-tagged WT or mutated Geminin in combination with other constructs and harvested for IP and WB. Source data are provided in this paper or Mendeley database (10.17632/8n7xt5rkhc.1). Similar results for (**a**), (**b**), (**c**), (**e**), and (**f**) panels were obtained in two independent experiments.

expression of SPOP mutants F102C and F133V had no effect on the interaction between ectopically expressed Geminin and Cdt1 (Supplementary Fig. 3a). Similarly, stable expression of F133V failed to affect the Geminin-Cdt1 interaction in BPH1 and DU145 cell lines at endogenous level (Supplementary Fig. 3b). Mutation of Geminin ubiquitination sites Lys100 and Lys127 also had no effect on Geminin binding to Cdt1 (Supplementary Fig. 3c). Since binding of the MCM complex to Cdt1 is critical for regulation of DNA replication firing[20,21], we further investigated whether SPOP-mediated Geminin ubiquitination impacts MCM complex access to Cdt1. As expected, Geminin knockdown increased Cdt1 binding with pre-RC proteins such as MCM2, CDC6, and ORC2, and this process was reversed by expression of

shRNA-resistant WT Geminin but not the K100R/K127R mutant (Fig. 4a and Supplementary Fig. 3d). Similar to the effect of the ubiquitination-resistant mutant of Geminin, expression of cancer-derived SPOP mutants also increased Cdt1 binding to MCM2, CDC6, and ORC2 but had no influence on Geminin binding to Cdt1 (Fig. 4b and Supplementary Fig. 3e).

Current structural knowledge of the replication origin and origin recognition complex formation is mostly derived from studies in budding yeast[46,47]. A single-particle cryo-EM structure of a pre-insertion loading intermediate of ORC-Cdc6-Cdt1-MCM (OCCM) complex from yeast was recently obtained by truncating the C-terminal WD domain of Mcm6 (PDB 6WGG)[47]. This structural intermediate, which precedes OCCM formation during

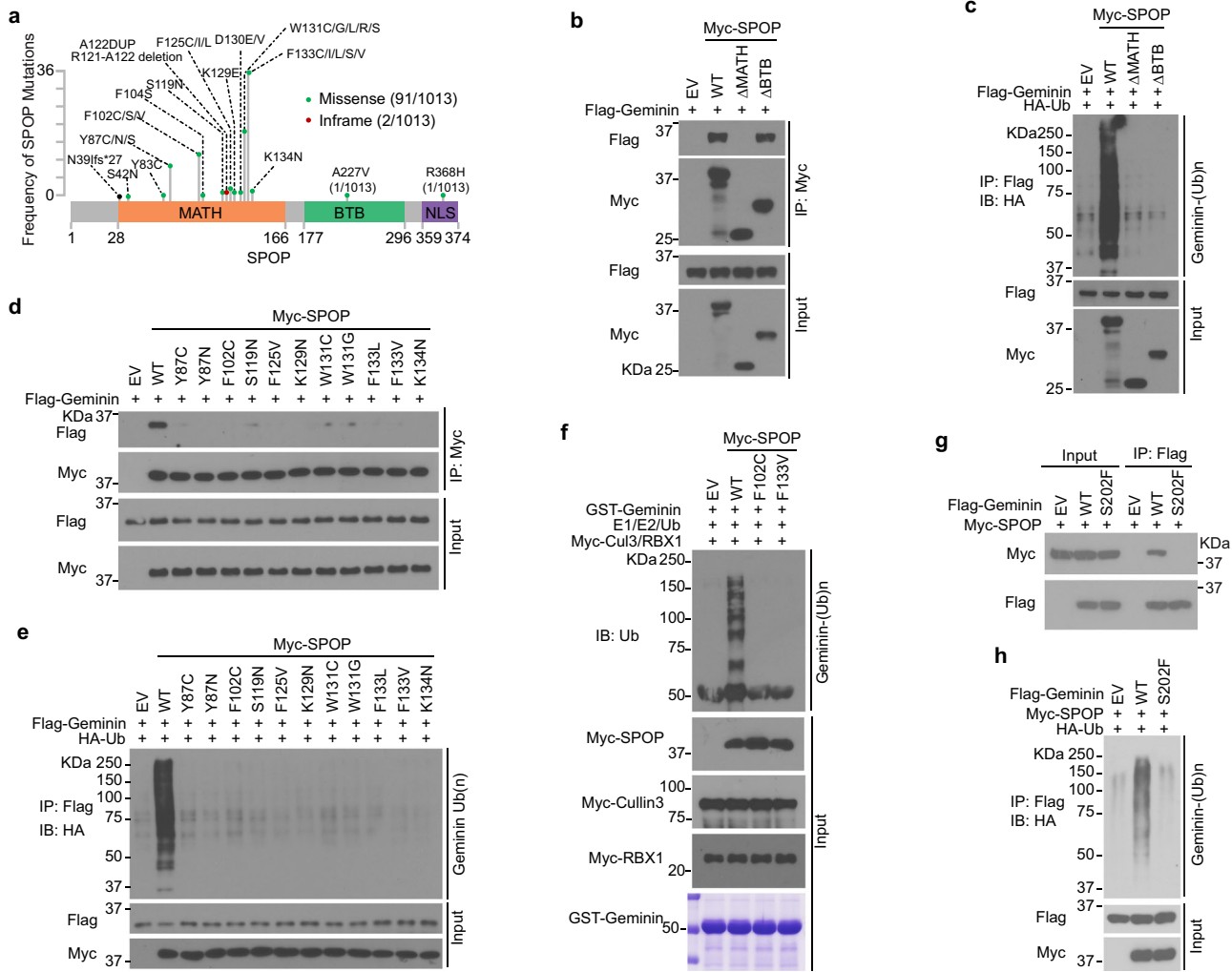

**Fig. 3 Prostate cancer-derived SPOP mutants fail to promote Geminin poly-ubiquitination. a** Schematic of domain organization of SPOP and SPOP mutations detected in 1013 cases of prostate cancers[30]. **b** 293T cells transfected with Flag-Geminin in combination with empty vector (EV), SPOP WT, or mutants were harvested for co-IP and WB with indicated antibodies. **c** 293T cells were transfected with the indicated plasmids and harvested for IP to detect the ubiquitination of Geminin. **d** 293T cells were co-transfected with Flag-Geminin and SPOP WT or mutants and harvested for co-IP and WB. **e** 293T cells were co-transfected with Flag-Geminin and SPOP WT or mutants and harvested for co-IP and WB to detect Geminin poly-ubiquitination. **f** In vitro ubiquitination assay was performed by incubating the reconstituted E1 and E2 enzymes with CUL3/RBX1, Myc-SPOP WT, or mutants immunoprecipitated from 293T cells and GST-Geminin purified from *E. coli*. **g, h** Co-IP analysis of indicated proteins and ubiquitination in 293T cells transiently transfected with Flag-Geminin-WT or S202F mutant. Source data are provided in this paper or Mendeley database (10.17632/8n7xt5rkhc.1). Similar results for (**b**–**h**) panels were obtained in two independent experiments.

the helicase loading process, shows how Cdt1 associates with the MCM complex. Despite low sequence similarities, Cdt1 from human (hCdt1) and yeast (yCdt1) both have conserved winged-helix domains (WHD) in their middle (M-WHD) and C-terminal (C-WHD) regions (Supplementary Fig. 3f)[48]. Because of these structural similarities, we built a model by using the crystal structure of the Geminin-bound M-WHD region of hCdt1 (PDB 2WVR)[26] to replace the corresponding yCdt1 region in the aforementioned OCCM intermediate complex (PDB 6WGG)[47]. The hCdt1 and yCdt1 M-WHD structures can be overlaid without any steric interference on the MCM complex (Supplementary Fig. 3g). The C-terminal part of bound Geminin, however, creates a steric hindrance. Molecular dynamics simulations based on the hCdt1-Geminin crystal structure revealed large amplitude motions of the C-terminal helical regions of the Geminin homodimer construct used for crystallography (Supplementary Fig. 3h). Taking this flexibility into account, we adjusted and optimized the conformation of Geminin C-terminal regions

to eliminate any steric clash. Our structural model for Geminin-bound OCCM (Fig. 4c) predicts that K27-linked poly-ubiquitination at Geminin Lys127 would cause a steric clash with the MCM complex that would displace Cdt1 from MCM. This analysis provides a plausible explanation as to how SPOP-mediated ubiquitination of Geminin affects the Cdt1-MCM interaction, while not directly affecting the Geminin-Cdt1 interaction. In agreement with our model in Fig. 4c, co-IP and mass spectrometry assays showed that Geminin could pull down MCM proteins and that SPOP-mediated poly-ubiquitination of Geminin reduced Geminin-MCM complex formation (Supplementary Fig. 3i–k). In addition, in vitro translated MCM2 protein showed no direct binding to GST-Geminin purified from bacteria (Supplementary Fig. 3l), indicating that Geminin indirectly associates with MCM proteins.

We further performed fluorescence-activated cell sorting (FACS) analysis to determine the effect of SPOP ubiquitination of Geminin on cell re-replication. Geminin knockdown increased

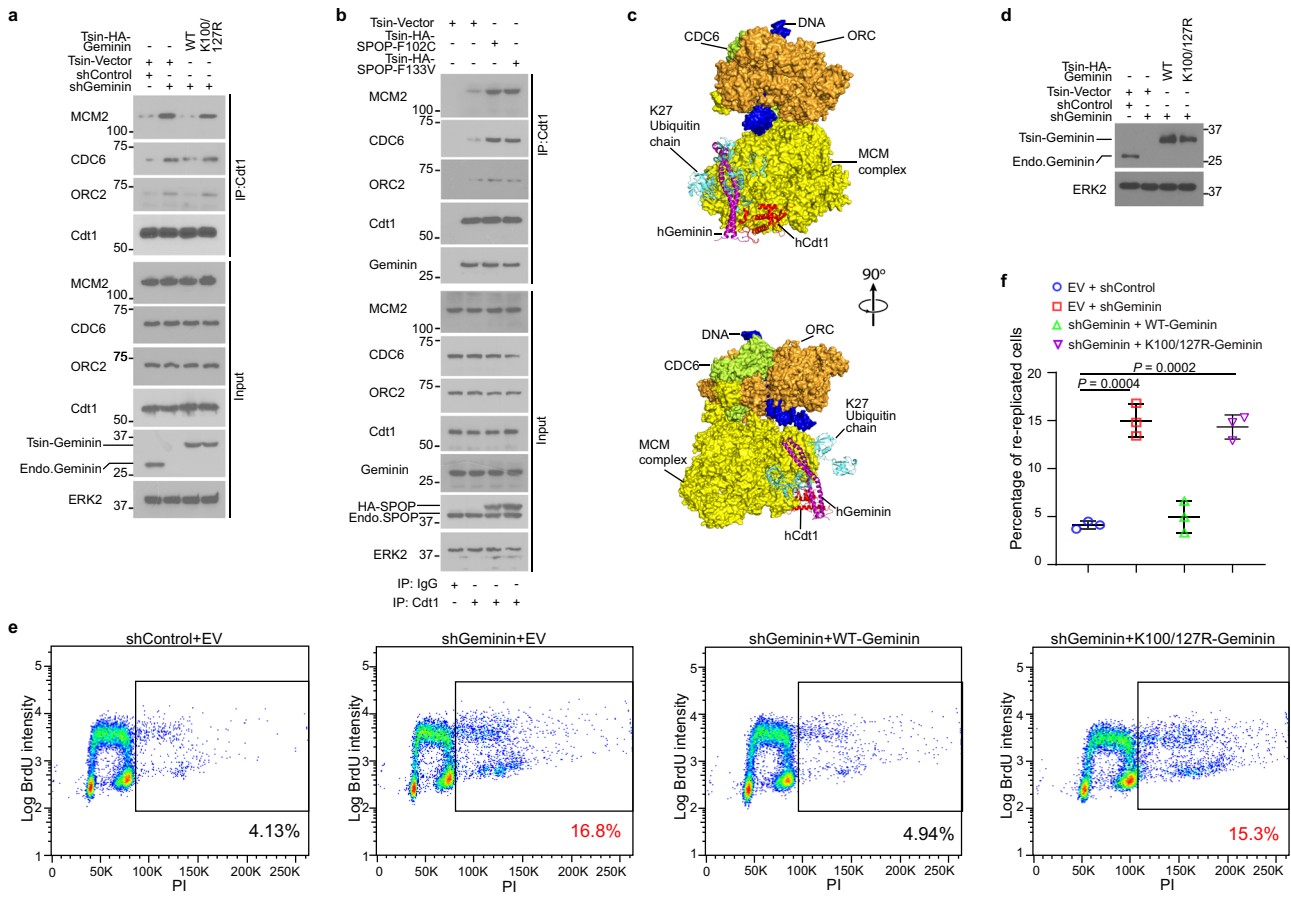

**Fig. 4 SPOP-mediated poly-ubiquitination of Geminin inhibits MCM proteins binding to Cdt1. a** PC-3 cells were infected with lentivirus expressing indicated shRNAs or WT or mutated Geminin and harvested for co-IP and WB analysis. **b** PC-3 cells were infected with lentivirus expressing empty vector or different SPOP mutants and harvested for co-IP and WB. **c** The structure of human Cdt1-Geminin is overlaid to that of yeast Cdt1-MCM after conformational optimization. A modeled K27-linked poly-ubiquitin chain attached to Geminin Lys127 (light blue) is incompatible with Cdt1-Geminin binding to the MCM complex. **d**–**f** PC-3 cells were infected with lentivirus expressing control shRNA or Geminin-specific shRNA in combination with empty vector (EV), Geminin WT, or K100/127R mutant. Cells were pulse-labeled with 30 μM BrdU prior to WB analysis (**d**), FACS analysis (**e**), or quantification (**f**). Data are presented as the mean ± SD of three independent experiments. Two-tailed unpaired Student's *t*-test; ***$P < 0.001$. Source data are provided in this paper or Mendeley database (10.17632/8n7xt5rkhc.1). Similar results for (**a**), (**b**), and (**d**) panels were obtained in three independent experiments.

re-replication in PC-3 cells and the increase was reversed by restored expression of shRNA-resistant WT Geminin, but not by the K100R/K127R double mutant (Fig. 4d–f). Together, these data suggest that SPOP-mediated Geminin poly-ubiquitination is important for Geminin inhibition of the replication function of Cdt1.

**Prostate cancer-derived SPOP mutants increase replication origin firing, re-replication, and genome instability, especially upon ATR inhibition.** The Cdt1-inhibitory function of Geminin is activated at the S and early G2 phases of the cell cycle to prevent the assembly of pre-RC[16,19,26,49]. We sought to determine whether SPOP regulates Geminin poly-ubiquitination in a cell-cycle-dependent manner. To this end, PC-3 cells were synchronized through thymidine and L-mimosine double block (Fig. 5a). We observed a slight increase in a population with DNA content over tetraploid (>4N) in SPOP-F133V mutant cells at S and early G2 phases (Fig. 5a). SPOP protein oscillated during the cell cycle with increased expression at G1 and S phases in empty vector (EV)-expressing control cells (Fig. 5b). Geminin protein level also started to increase from S phase after a decrease at G1 phase in EV cells (Fig. 5b), similar to what was previously reported[15,49–52]. However, when chromatin binding of pre-RC proteins such as MCM2 began to decrease after completion of replication and

entry into early G2 phase in control cells, MCM2 association with chromatin remained high at S phase and early G2 phase in SPOP-F133V mutant cells (Fig. 5b, c), indicating that the replication machinery remains active at this stage of the cell cycle in SPOP F133V mutant cells. Both SPOP and Geminin protein levels fluctuated during the cell cycle and plateaued at S and early G2 phases (Fig. 5b), implying that these two proteins might work in concert at these stages of the cell cycle.

To determine the impact of cell cycle-dependent oscillation of SPOP protein on Geminin poly-ubiquitination, we synchronized PC-3 cells expressing HA-tagged K27-only ubiquitin (Ub) (HA-Ub-K27) by nocodazole (Supplementary Fig. 4a). A dramatic increase in Geminin K27-linked poly-ubiquitination was observed in control cells at S and early G2 phases compared to cells at other phases (Fig. 5d). Notably, both the substrate (Geminin) and the E3 ligase (SPOP) also reached the highest protein level at these two phases in control PC-3 cells (Fig. 5b). However, the cell-cycle-dependent K27-linkage poly-ubiquitination of Geminin was completely abolished in SPOP-F133V mutant-expressing PC-3 cells even though Geminin protein level retained a similar oscillation pattern as in control cells (Fig. 5d). Notably, the total poly-ubiquitination of Geminin was also decreased at S and early G2 phases in F133V mutant cells transfected with WT ubiquitin (Supplementary Fig. 4b). Furthermore, FACS analysis with BrdU-

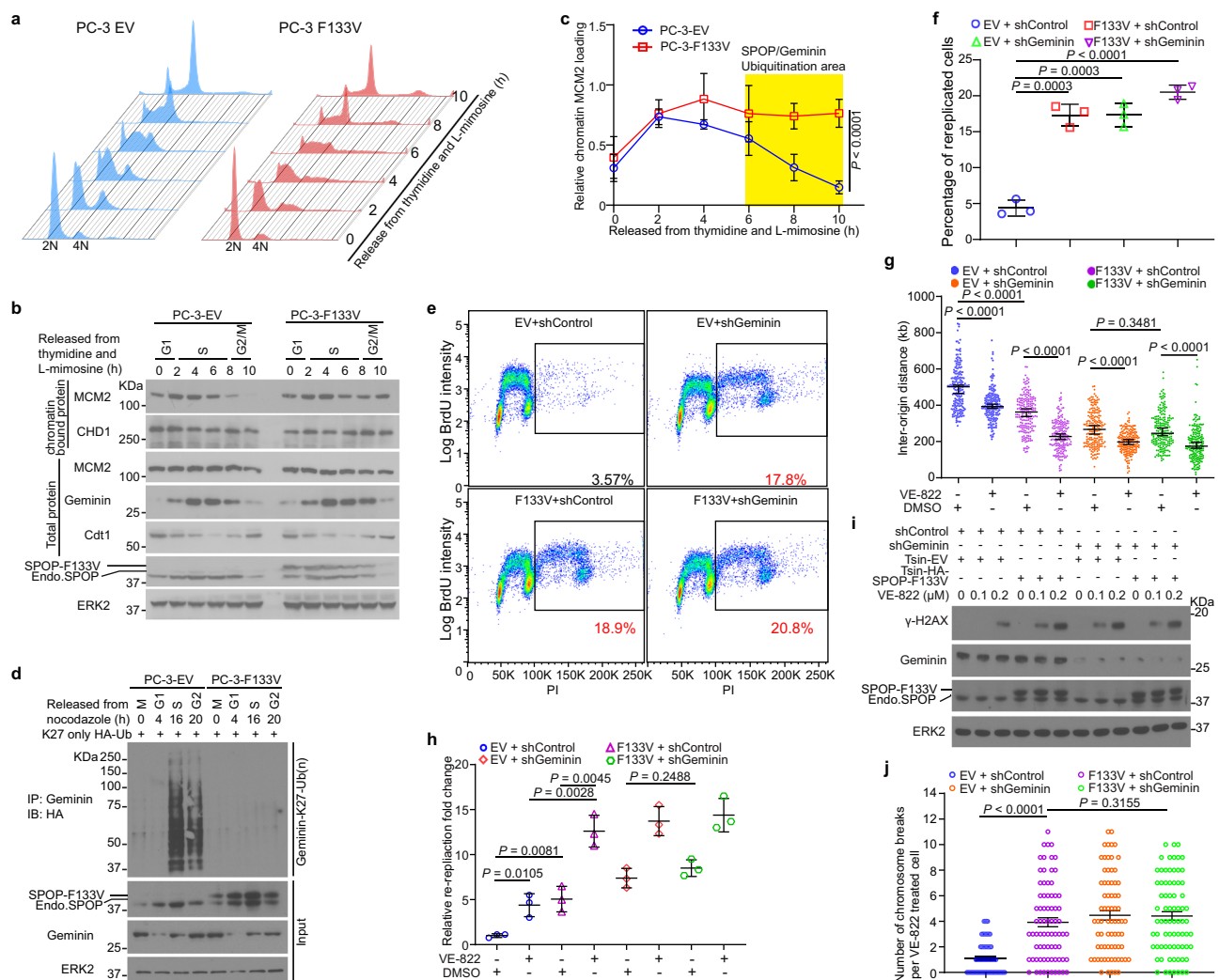

**Fig. 5 Prostate cancer-derived SPOP mutants increase replication origin firing, re-replication, and genome instability, especially upon ATR inhibition.**
**a** PC-3 cells stably expressing empty vector (EV) and SPOP-F133V mutant were synchronized by thymidine and L-mimosine double blockade. Cells were released and harvested at the indicated time points for FACS-based cell cycle analysis. **b** PC-3 cells expressing EV and SPOP F133V were synchronized as in (**a**) and harvested for WB. Similar results were obtained in three independent experiments. **c** Protein bands for chromatin-associated MCM2 from the experiment shown in (**b**) and the other two repeats were quantified using ImageJ and normalized to total MCM2 protein level. Data are presented as the mean ± SD ($n = 3$). The $P$ value was calculated by two-way ANOVA analysis. **d** EV- and F133V-expressing PC-3 cells transfected with HA-Ub-K27-only were synchronized with nocodazole and released at different time points and harvested for IP and WB. **e, f** PC-3 cells infected with lentivirus expressing EV or SPOP F133V in combination with control or Geminin-specific shRNAs. Cells were pulse-labeled with 30 μM BrdU for 30 min and harvested for FACS analysis (**e**) and quantitative data are shown in (**f**). Data are shown as the mean ± SD of three independent experiments. Two-tailed unpaired Student's t-test. **g** PC-3 cells infected with lentivirus expressing empty vector (EV) or F133V in combination with control or Geminin-specific shRNAs were treated with DMSO or 100 nM VE-822 for 8 h followed by DNA fiber assay. Inter-origin distances between 200 replication firing origins were measured ($n = 200$). Data are shown as the mean ± SD of three independent experiments. Two-tailed unpaired Student's t-test. **h** Stable PC-3 cells as indicated were treated with DMSO or 100 nM VE-822 for 8 h followed by DNA fiber assay. DNA re-replication was quantified from 200 DNA tracks and relative re-replication fold changes are presented. Data are shown as the mean ± SD of three independent experiments ($n = 3$). Two-tailed unpaired Student's t-test. **i** PC-3 cells infected with lentivirus expressing the indicated shRNAs or EV or SPOP F133V were treated with DMSO or increased doses of VE-822 for 24 h and harvested for WB with indicated antibodies. **j** Stable PC-3 cells as indicated were treated with VE-822 (100 nM) for 24 h. Cells were harvested for karyotyping. Quantification of chromosome breaks per cell are shown. More than 70 metaphases from four biological replicates were counted. Data are mean ± SEM. Two-tailed unpaired Student's t-test. Source data are provided in this paper or Mendeley database (10.17632/8n7xt5rkhc.1). Similar results for (**b**), (**d**), and (**i**) panels were obtained in two independent experiments.

labeled cells showed that SPOP-F133V expression resulted in an increase of re-replicated cells (>4N) in PC-3 cells (Fig. 5e, f and Supplementary Fig. 4c). A similar result was obtained in Geminin knockdown cells regardless of SPOP-F133V expression (Fig. 5e, f). These data demonstrate that SPOP mutant cells undergo an aberrant replication process including overloading of the pre-RC proteins onto chromatin (Fig. 5b, c) and ultimately DNA re-replication (Fig. 5e, f).

To further assess the effect of SPOP mutation on re-replication, we employed DNA fiber assay to gauge abnormal replication. The ATR-CHK1 pathway plays a key role in suppressing dormant origin over-firing and preventing RPA depletion in nuclear pool which protects DNA replication forks from collapse at S phase[53–55]. We included VE-822, an ATR inhibitor in our DNA fiber assay to determine whether ATR inhibition has any additional effects on DNA re-replication in SPOP mutant cells.

As a positive control, we found that the distance between two adjacent origins (inter-origin distance) was decreased in cells treated with VE-822 (Fig. 5g and Supplementary Fig. 4d). However, this phenomenon was also observed in Geminin knockdown or SPOP-F133V mutant-expressing cells without VE-822 treatment (Fig. 5g). Importantly, the inter-origin distance in F133V and/or Geminin knockdown cells was lower than that in control cells and was further reduced after co-treatment of VE-822 (Fig. 5g). The decreased inter-origin distance indicates that there is increased origin firing in SPOP-mutated cells, providing a plausible explanation as to why there is increased binding of pre-RC proteins to chromatin (Fig. 5b). DNA re-replication also increased by several folds in PC-3 cells expressing SPOP mutant or Geminin shRNA compared to control cells (Fig. 5h and Supplementary Fig. 4e), and their effects were further enhanced by ATR inhibitor treatment (Fig. 5h). Both increased replication stress and re-replication burden sensitize ATR inhibition-caused replication catastrophe and double-strand breaks and ultimately lead to cell death[54]. We assessed whether this would also be the case in SPOP mutant cells. As expected, treatment with ATR inhibitor alone increased the level of γ-H2AX in control PC-3 cells and this effect was largely enhanced by expression of SPOP mutant F133V or Geminin knockdown (Fig. 5i). However, combination of SPOP-F133V expression and Geminin knockdown failed to induce more DNA breaks than each condition alone in PC-3 cells treated with ATR inhibitor (Fig. 5i). These data suggest that SPOP and Geminin work in the same pathway in prohibition of ATR inhibition-caused replication catastrophe and double-strand breaks. We also checked the numbers of intra-chromosomal breaks per cell after these groups of cells were treated or not with ATR inhibitor (Fig. 5j and Supplementary Fig. 4e, f). The results were consistent with changes in γ-H2AX level (Fig. 5i). Therefore, prostate cancer-associated SPOP mutation F133V impairs DNA re-replication checkpoint, promotes chromosomal instability, and leads to replication catastrophe when ATR is inhibited.

**SPOP mutant cells are hypersensitive to ATR inhibition**. Because SPOP mutant cells acquire a marked increase in replication stress and re-replication burden (Fig. 5g, h), and encounter replication catastrophe upon ATR inhibition (Fig. 5i, j), we hypothesized that SPOP-mutated cells are hypersensitive to ATR inhibitors due to replication catastrophe. To test this hypothesis, we measured the viability of SPOP mutant-expressing BPH1, C4-2, 22RV1, DU145, and PC-3 cells treated with two different ATR inhibitors (AZD6738 and VE-822). A dose surviving assay demonstrated that expression of SPOP mutant F133V in all five cell lines resulted in decreased IC50 of both inhibitors compared with EV control cell lines (Fig. 6a and Supplementary Fig. 5a–c). ATR knockdown markedly inhibited growth of SPOP mutant cells with formation of fewer and smaller colonies while ATR knockdown in control cells only slightly decreased the size and number of colonies (Fig. 6b, c and Supplementary Fig. 5d, e). We also examined the effect of ATR inhibition by VE-822 in a clinically-relevant setting by using the SPOP Q165P mutant patient-derived xenograft (PDX) model established from a prostate cancer metastatic lesion[45]. Similar to the other SPOP mutants, such as F102C and F133V, Q165P lost the ability to bind to and ubiquitinate Geminin (Supplementary Fig. 5f, g). Similar to the results in F133V-expressing cells, both Q165P mutant-expressing DU145 and PC-3 cells had much lower IC50 doses of VE-822 (Fig. 6d, e and Supplementary Fig. 5h) and VE-822 treatment resulted in much smaller colony numbers compared to vehicle-treated cells (Fig. 6f–i). We established organoids from Q165P PDX tumors. We showed that the diameter of mock-treated SPOP Q165P organoids was much larger than that of the WT counterpart, but the diameter of SPOP Q165P organoids was much smaller than that of the control organoids when they were treated with VE-822 (Fig. 6j, k). We also treated SPOP-WT and Q165P PDX tumors with VE-822 in mice. Similar to the finding in organoid culture, SPOP Q165P mutant PDX tumors were more sensitive to VE-822 compared to SPOP-WT PDX tumors (Fig. 6l, m). Our data demonstrate that SPOP-mutated prostate cancer cells are hypersensitive to ATR inhibition in vitro and in vivo.

## Discussion

Although SPOP is broadly recognized as a tumor suppressor in prostate cancer and SPOP mutations are associated with high frequency of genomic rearrangements, the molecular mechanisms by which SPOP mutations promote genome instability remain poorly understood. In the present study, we identify a role for SPOP in guarding DNA replication and chromosomal stability. Based upon our findings, we envision a model wherein SPOP mutant cells have increased replication origin firing and re-replication burden, causing replication catastrophe and cell death upon ATR inhibition (Fig. 7). Therefore, our work uncovers a previously unrecognized tumor suppressor role of SPOP in preventing DNA from over-replication and genome instability.

Geminin plays a pivotal role in regulating replication licensing, ensuring only one replicate of DNA per cell cycle. Accordingly, the expression level of Geminin protein is low or undetectable in G1 and surges through S to early G2 phases of the cell cycle. However, Geminin activity does not entirely correlate with its expression level, implying that there exist additional mechanisms regulating Geminin activity beyond its protein level. Intriguingly, a previous study identified a $^{98}$YWK$^{100}$ motif in Geminin that is critical for Geminin to prevent DNA replication from over-firing, although the underlying mechanism was unexplored[56]. Our data reveal that the Lys100 in the $^{98}$YWK$^{100}$ motif is one of the SPOP ubiquitination sites. Most importantly, we show that while Geminin K100R mutant retains its ability to bind to Cdt1, which is similar to the ability of previously reported $^{98}$AWA$^{100}$ mutant[56], SPOP-mediated ubiquitination of Geminin K100R mutant is substantially reduced. Thus, our findings provide a mechanistic explanation for the critical function of Geminin $^{98}$YWK$^{100}$ motif and its poly-ubiquitination in preventing DNA over-replication.

It has long been suggested that different replication licensing statuses of Geminin/Cdt1 exist[24–26,57]. However, how the equilibrium between a licensing permissive to a licensing inhibitory status shifts during the cell cycle remains an open question. Our work suggests that SPOP-dependent Geminin poly-ubiquitination indirectly blocks MCM access to Cdt1 and that this effect might be achieved through steric hindrance as our structural model predicts. This posttranslational modification of Geminin could be a functional switch for the Geminin-Cdt1 complex. We acknowledge that other mechanisms are possible, but our proposed model, consistent with experimental results, provides the simplest possible explanation for how SPOP regulates DNA replication via Geminin ubiquitination (Fig. 7b). Among other explanations, one could be a general conformational change in Geminin-Cdt1-MCM complex caused by SPOP-induced poly-ubiquitination of Geminin. Consistent with the finding that K27-linked poly-ubiquitination of Geminin does not cause protein degradation and that such modification is reversible, our work also suggests that oscillations in SPOP protein levels during the cell cycle drive fluctuations in Geminin poly-ubiquitination, which allows re-formation of the permissive status for the replication firing in the next cell cycle.

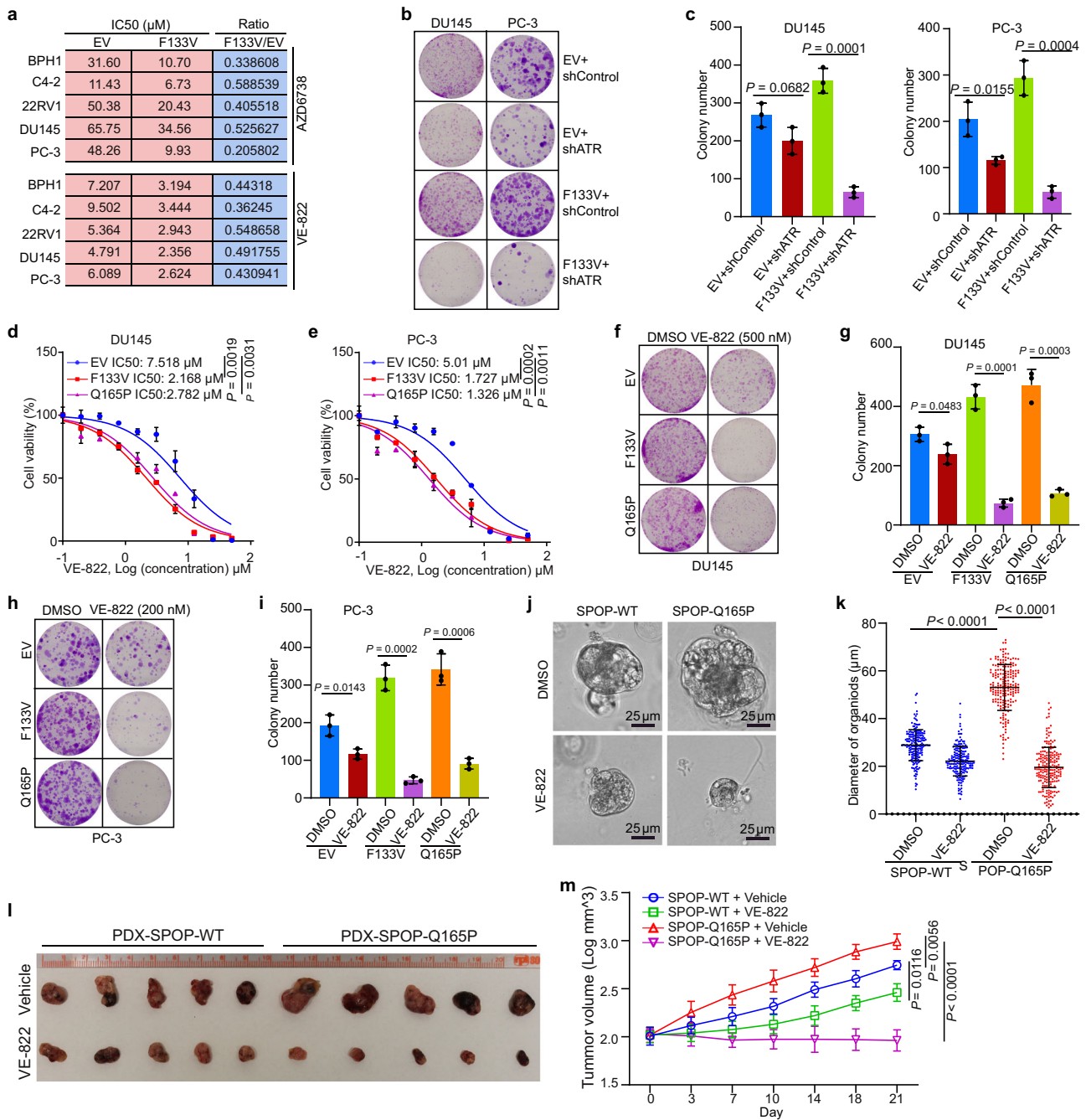

**Fig. 6 SPOP mutant cells are sensitive to ATR pathway inhibition. a** IC50 analysis of two ATR inhibitors in five prostate cell lines expressing EV or SPOP-F133V mutant. **b, c** Colony formation assays were performed in DU145 and PC-3 cell lines infected with lentivirus expressing control or ATR-specific shRNA or empty vector (EV) or SPOP mutant F133V. The number of colonies was counted. Representative colonies are shown in (**b**) with quantification data shown in (**c**). Data are presented as the mean ± SD of three independent experiments. Two-tailed unpaired Student's t-test. **d, e** Dose-response survival curves of EV, SPOP F133V, and SPOP Q165P cells exposed to increasing concentrations of VE-822 in DU145 (**d**) and PC-3 (**e**) cells. Data are shown as the mean ± SD of three independent experiments (n = 3 replicates/group). Two-tailed unpaired Student's t-test. **f-i** Colony formation assay was performed in DU145 (**f, g**) and PC-3 (**h, i**) cell lines treated with DMSO or VE-822. The number of colonies was counted. Representative colonies are shown in (**f, g**) with quantification data shown in (**h, i**). Data are shown as the mean ± SD of three independent experiments (n = 3). Two-tailed unpaired Student's t-test. **j, k** SPOP WT and Q165P organoid lines derived from Q165P PDX tumors were cultured for 5 days, followed by treatment with DMSO or VE-822 (200 nM) for five more days. The representative images of organoids after the treatment are shown in (**j**) and the quantified data of the organoid diameter are shown in (**k**). All data are shown as mean ± SD (n = 200). The P value was calculated using unpaired two-tailed Student's t-test. **l, m** SPOP-WT or SPOP Q165P PDX tumors were transplanted subcutaneously into SCID mice and treated with VE-822 (60 mg/kg, 5 times weekly by oral gavage) or vehicle. Mice were treated for 3 weeks and then sacrificed. Xenograft tumors were isolated and are shown in (**l**). Log of quantified volumes of the tumors from (**l**) (n = 5) are shown in (**m**). All data are shown as mean ± SD. The P value was calculated by two-way ANOVA analysis. Source data are provided in this paper.

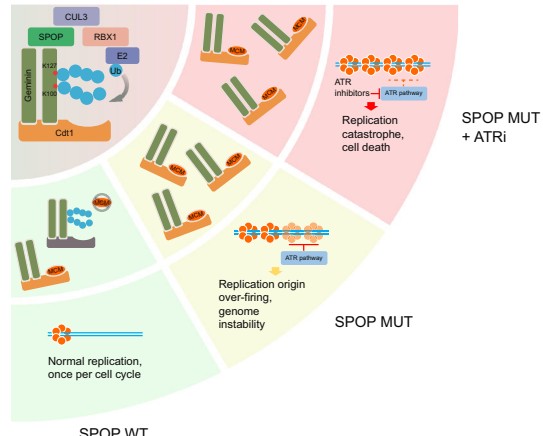

**Fig. 7 A hypothetical model depicting how SPOP mutation impairs DNA replication and sensitizes cancer cells to ATR inhibition. a** SPOP-CUL3-RBX1 functions as an E3 ubiquitin ligase that binds to Geminin and catalyzes K27-linked poly-ubiquitination of Geminin at Lys100 and Lys127. **b** SPOP-dependent Geminin poly-ubiquitination blocks Cdt1 binding to MCM proteins. Poly-ubiquitination keeps the licensing-competent state in check and prevents DNA replication over-firing. **c** Cancer-associated SPOP mutations impair DNA replication surveillance and cause replication origin over-firing and re-replication. **d** SPOP mutations trigger replication catastrophe and cancer cell death upon ATR inhibition.

Whole-genome and exome sequencing of cancer patient samples have shown that *SPOP* is the most frequently mutated gene in primary prostate cancer[27,28], suggesting that patients with SPOP mutations represent an important subtype of prostate cancer. SPOP mutations typically occur in a heterozygous state with a retained wild-type allele and are able to dysregulate known substrates in a dominant-negative manner[32,33,58]. Our finding also confirmed the dominant-negative effect of SPOP mutations. Almost all mutants lost the ability to bind and ubiquitinate Geminin even though endogenous SPOP is present (Fig. 3d, e). Increasing evidence indicates that SPOP targets a large spectrum of protein substrates for degradation, which implies that dysregulation of different downstream signaling pathways would require different therapeutic strategies for SPOP-mutated patients. Indeed, it has been shown that cells expressing prostate cancer-derived SPOP mutants are resistant to BET inhibitor due to elevated expression of BET family proteins BRD2, BRD3, and BRD4[32,33]. In contrast, given that SPOP mutants are unable to activate the inhibitory function of Geminin in constraining DNA replication over-firing (Fig. 7c), SPOP mutation triggers replication catastrophe upon ATR inhibition (Fig. 7d). This model is further supported by our finding that SPOP-mutated prostate cancer cells are hypersensitive to ATR inhibition. Thus, our findings shed new light on the development of new therapeutics for patients with SPOP-mutated prostate cancer.

In conclusion, we demonstrate that SPOP plays an important role in ensuring the normal process of DNA replication by controlling Geminin poly-ubiquitination and the switch of the Geminin/Cdt1 complex from the replication licensing-competent to the licensing-defective state. We further show that SPOP protein expression plateaued at S and early G2 phases, thereby triggering the highest level of poly-ubiquitination of Geminin and prevention of aberrant DNA re-replication at these stages of the cell cycle in normal cells. We also reveal that mutations in SPOP result in Geminin inactivation and undesired replication over-firing and re-replication. Our finding that SPOP mutation triggers replication catastrophe, massive DNA breaks, and cell death upon ATR inhibition highlights that prostate cancers harboring SPOP mutations may be susceptible to treatment with ATR inhibitors.

## Methods

**Cell lines, cell culture, and transfection.** The immortalized human embryonic kidney cell line 293T and prostate cancer cell lines PC-3, DU145, and 22RV1 were purchased from ATCC (Manassas, VA). C4-2 cells were purchased from Uro Corporation (Oklahoma City, OK). BPH1 cells were kindly provided by Dr. Simon Hayward. 293T cells were cultured in Dulbecco's modified Eagle's medium (DMEM) supplemented with 10% of FBS (Thermo Fisher Scientific). BPH1, DU145, PC-3, C4-2, and 22RV1 cells were cultured in RPMI 1640 medium supplemented with 10% FBS. The cells were maintained in a 37 °C humidified incubator supplied with 5% $CO_2$. Transient transfection was performed by Lipofectamine 2000 (Thermo Fisher Scientific). Lentiviral shRNA constructs were transfected using Calcium Phosphate protocol. pTsin-HA-SPOP-F133V mutant expression or lenticrisprV2-SPOP and virus packing constructs were transfected into 293T cells. Virus supernatant was collected 48 h after transfection. Prostate cells were infected with viral supernatant in the presence of polybrene (8 μg/ml) and were then selected in growth media containing 1.5 μg/ml puromycin. All the cell lines used have been tested and authenticated by karyotyping and prostate cancer cell lines have also been authenticated by examining SPOP mutation status. Plasmocin (InvivoGen) was added to cell culture media to prevent mycoplasma contamination. Mycoplasma contamination was tested regularly using Lookout Mycoplasma PCR Detection Kit from Sigma-Aldrich. The antibodies, reagents, primers, and other resources are listed in Supplementary Table 4.

**Antibody information.** Primary antibodies used were Geminin (Santa Cruz, # sc-74456, 1:500 (antibody dilution, below is the same)), Cdt1 (Santa Cruz, # sc-365305, 1:500), MCM2 (Santa Cruz, # sc-373702, 1:500), MCM3 (Santa Cruz, # sc-166940, 1:500), MCM4 (Santa Cruz, Cat# sc-28317, 1:500), MCM7 (Santa Cruz, # sc-9966, 1:500), Cdc6 (Santa Cruz, # sc-9964, 1:500), ORC2 (Santa Cruz, # sc-32734, 1:500), HA.11 (Covance, #MMS-101R, 1:1000), BRD4 (Abcam, # ab1228874, 1:1000), SPOP (Proteintech Group, # 16750-1-AP, 1:1000), Myc (Santa Cruz, # sc-40, 1:1000), Flag (Sigma, # F-3165, 1:1000), ERK2 (Santa Cruz, # sc-1647, 1:1000), Phospho Histone H2A.X (S139) (Cell Signaling, # 9718, 1:1000), BrdU (Abcam, # ab6326, 1:500), BrdU (BD Bioscience, # 347580, 1:500). Second antibodies were Rabbit IgG (H + L) Alexa Fluor 594 (Thermo Fisher, # A11037, 1:500), Rabbit IgG (Jackson ImmunoResearch, # 211-032-171, 1:5000), Mouse IgG (Jackson ImmunoResearch, #115-035-174, 1:5000), Mouse IgG (H + L) Alexa Fluor 488 (Thermo Fisher, # A11029, 1:500), Rat IgG (H + L) Alexa Fluor 488 (Life Technologies, # A-11006, 1:500), Mouse-IgGκ BP-FITC (Santa Cruz, # sc-516140, 1:500).

**Protein and peptide preparation.** $^{15}$N-labeled SPOP MATH (amino acids 28–166) harboring a hexahistidine tag cleavable with PreScission protease was expressed in *E. coli* BL21 (DE3) cells grown in $^{15}$NH$_4$Cl-enriched M9 media. The cells were grown at 37 °C to an OD$_{600}$ of 0.6 and then at 15 °C overnight after addition of isopropyl-β-D-thiogalactoside (IPTG, final concentration 0.5 mM). The harvested cells were lysed using an Avestin Emulsiflex C5 cell disruptor (Avestin Inc., Ottawa, Canada). The protein was purified by nickel affinity chromatography (QIAGEN) and incubated with PreScission protease overnight at 4 °C to cleave the hexahistidine tag. The protein was further purified by size exclusion chromatography on a preparative Superdex 75 16/600 column (GE Healthcare). Using the same protocol, we also purified non-labeled SPOP MATH (D140G mutant[34]) for X-ray crystallography. Non-labeled Geminin wild-type SBC ($^{195}$AEGTVSSST-DAKPCI$^{209}$) and alanine mutant SBC ($^{195}$AEGTVAAAADAKPCI$^{209}$) synthetic peptides were purified by reversed-phase chromatography using a Jupiter 5 μm C18 300 preparative column (Phenomenex). Stocks of the peptides (15 mM) were prepared in the NMR buffer.

**Nuclear magnetic resonance spectroscopy.** For the NMR spectroscopy experiments, SPOP MATH was in a final buffer (NMR buffer) containing 10 mM Na$_2$HPO$_4$, 1.76 mM KH$_2$PO$_4$, 50 mM NaCl, 2.7 mM KCl, 5 mM dithiothreitol, 90% H$_2$O/10% D$_2$O, pH 6.0. SPOP-MATH resonance assignments were taken from the Biological Magnetic Resonance Bank entry 26629[59] and confirmed by recording a 3D $^{15}$N-NOESY spectrum for the $^{15}$N-labeled protein. $^{15}$N-labeled SPOP MATH, at a concentration of 0.2 mM, was titrated with up to 3-fold molar excess of each of the Geminin peptides. $^{1}$H-$^{15}$N HSQC NMR spectra were collected for SPOP MATH, free and bound to each peptide, at 298 K using a 700 MHz Bruker AVANCE III spectrometer equipped with a cryoprobe. The NMR data were processed with NMRPipe[60] and analyzed with NMRViewJ (OneMoon Scientific, Inc.).

**X-ray crystallography.** Prior to crystallization, purified SPOP MATH (D140G mutant[34]) in 50 mM sodium phosphate buffer, pH 7.5, 300 mM NaCl was transferred to 20 mM Tris-HCl, pH 7.6, 50 mM NaCl, and 5 mM dithiothreitol using a Centricon centrifugal filter (Sigma-Aldrich). For crystallization, the SPOP-MATH-Geminin peptide sample contained 1.1 mM SPOP-MATH domain (D140G mutant) and 5.5 mM Geminin peptide in 20 mM Tris-HCl, pH 7.6, 50 mM NaCl, and 5 mM dithiothreitol. Crystals were grown at 22 °C using the sitting-drop method, mixing 1 μl of SPOP-MATH-Geminin complex and 1 μl of reservoir solution containing 0.2 M ammonium citrate dibasic and 20% (w/v) PEG 3350. Crystals appeared after 3 to 8 weeks. Crystals were cryoprotected with 25% (w/v) xylitol and snap-frozen in liquid nitrogen. Diffraction data were collected at the 19-

BM beamline of the Advanced Photon Source at Argonne National Laboratory, IL. The diffraction data were processed with HKL2000[61]. Starting phases were obtained by molecular replacement using PDB entry 6F8G[62]. The initial model was adjusted using COOT[63] and refined using PHENIX[64]. Because of our low resolution (3.4 Å) diffraction data, we used previously reported higher resolution structures of other SPOP-MATH-SBC complexes[34] to determine the polarity of the SBC-containing Geminin peptide we co-crystallized. Across different species and proteins, the SBC motif is highly conserved in sequence and in structure, with the nonpolar residue (Φ in the Φ-π-S-S/T-S/T SBC motif) surrounded by aromatic residues. All molecular representations were generated using PyMOL[65].

**Model building and molecular dynamics simulations**. The structure of human Geminin-Cdt1 complex (PDB code 2WVR)[26] was docked onto that of budding yeast OCCM (PDB code 6WGG)[47] by best-fit overlay of the structure of human Cdt1 with that of yeast Cdt1. The Geminin C-terminal helical region was altered to avoid structural clash with the structure of MCM2 using Coot[66]. This Geminin-OCCM structural model was optimized by molecular dynamics (MD) simulations. A model of a chain of three K27-linked ubiquitin molecules (PDB code 1UBQ)[67] attached to Geminin Lys127 via an isopeptide bond was generated using PyMol and optimized by MD simulations. The most abundant ubiquitin chain conformation was then grafted to Geminin Lys127 in the above Geminin-OCCM model.

The MD simulations, model optimizations, and data analyses were carried out using GROMACS (version 2020.2) with the all-atom CHARMM36 force field[68]. The proteins were in triclinic boxes and solvated with explicit TIP3P water molecules. Charges were neutralized with $Cl^-$ ions and NaCl was introduced at a concentration of 0.150 M. The systems were energy minimized using the steepest descent algorithm with a maximum force of 200 kJ·mol$^{-1}$·nm$^{-1}$. The temperature was then equilibrated via 200 ps of constant volume equilibration at 310 K using a velocity-rescaling thermostat[69]. This was followed by equilibration for 1.0 ns to a 1.0 bar constant pressure bath using the Berendsen weak coupling method. The above equilibration steps were performed with protein molecules position-restrained using a force of 1000 kJ·mol$^{-1}$·nm$^{-1}$. MD productions used periodic boundary conditions with a time step of 2.0 fs. The particle mesh Ewald method[66] with a Fourier grid spacing of 0.12 nm was used to calculate long-range electrostatic interactions. A leap-frog integration algorithm was used for the MD simulations[70] and bond lengths were restrained using the LINCS algorithm[71]. Trajectories were written every 20 ps.

**In vitro ubiquitination assay**. Myc-tagged CUL3, RBX1 expression vectors were co-transfected with empty vector, SPOP-WT, or two mutants in 293T cells and proteins immunoprecipitated from cell lysate were mixed with GST-Geminin purified from E. coli. The mixed protein was incubated with 5 μg Ub, 500 ng E1, 750 ng E2, 0.6 μl 100 mM ATP, 3 μl 10× ubiquitin reaction buffer (500 mM Tris-HCl pH 7.5, 50 mM KCl, 50 mM NaF, 50 mM MgCl$_2$, and 5 mM DTT), 3 μl 10× energy regeneration mix (200 mM creatine phosphate and 2 μg/μl creatine phosphokinase) and 3 μl 10× protease inhibitor cocktail at 30 °C for 2 h, followed by western blot (WB) analysis. The Ub, E1, and E2 were purchased from UBIQUIGENT.

**Cell proliferation assay**. BrdU/PI flow cytometry was used. Cells were first incubated with 30 μM BrdU for 30 min. After cells were digested with trypsin and washed in PBS, cells were resuspended in 70% ice-cold ethanol and stored at −20 °C. Before flow cytometry analysis, cells were washed three times in PBS and incubated in 2 N HCl and 0.5% Triton X-100 for 30 min at room temperature. Cells were next washed with washing buffer (PBS, 1% BSA and 0.2% Triton X-100), and consecutively blocked in blocking buffer (PBS, 5% BSA and 0.2% Triton X-100) without and with mouse anti-BrdU antibody for 1 h each. Cells were again washed three times and then incubated in blocking buffer with mouse-IgGκ BP-FITC for 30 min. Finally, cells were resuspended in 0.5 ml PBS containing 10 μg/ml RNase A and 20 μg/ml PI for flow cytometry analysis. Analysis was done using the FlowJo 10.4 analysis software (FlowJo LL).

**Chromatin fractionation assay**. Cells were washed in PBS and lysed in CSK buffer (10 mM HEPES pH 7.9, 100 mM NaCl, 300 mM sucrose, and 0.1% Triton X-100 and protease inhibitors) in ice for 20 min. After centrifugation at 11,200 × g for 10 min, the total lysate was collected. The pellet from the previous step was washed twice with CSK buffer and resuspended with DNase I and 300 mM NaCl at 25 °C for 30 min. After sonicating the resuspended pellet, the resulting lysate contained the chromatin fraction.

**In vivo ubiquitination assay**. For in vivo ubiquitination, 293T cells were transfected with plasmids for HA-Ub, Flag-Geminin, and other indicated constructs. Cells were harvested and lysed with lysis buffer (50 mM Tris-HCl, pH 7.5, 150 mM NaCl, 1% NP40, 0.5% sodium deoxycholate, and 1× protease inhibitor cocktail (PIC)). The lysate was subjected to co-immunoprecipitation using anti-Flag-conjugated agarose beads or Flag primary antibody plus protein A/G beads as described in co-IP assay.

**Cell synchronization**. PC-3 and 293T cells were treated with 2 mM thymidine for 24 h and released into regular culture medium for 3 h. After washing with PBS

three times, cells were released into regular medium for another 9 h, after which cells were blocked by L-mimosine (300 μM final concentration) for 24 h and released into regular medium. At the indicated time points after final release, cells were harvested for cell cycle profiling and western blot and co-IP analyses.

**DNA fiber assay**. DNA fiber assays were performed following published protocols[72,73]. Briefly, PC-3 cells were infected with lentivirus expressing empty vector (EV) or F133V in combination with control or Geminin-specific shRNAs and treated with DMSO or 100 nM VE-822 for 8 h. Cells were first-labeled with 25 mM IdU for 30 min, washed three times with PBS, and second-labeled with 250 mM CldU for 1 h. Labeled cells were harvested, and DNA fibers were spread by gravity. Primary antibody dilutions used were mouse anti-BrdU 1/20 (for IdU) and rat anti-BrdU (for CldU) 1/100. Images of well-spread DNA fibers were acquired using an LSM700 confocal microscope with 100× oil immersion objective (Carl Zeiss). Analysis of double-labeled replication forks was performed manually using LSM ZEN software (Carl Zeiss).

**Sample preparation for liquid chromatography-mass spectrometry (LC-MS)**. For LC-MS analysis of SPOP interactome, 293T cells were transfected with S, Flag, and Biotin-binding-protein-(streptavidin)-binding-peptide (SFB) triple-tagged backbone vector or SFB-SPOP. Twenty-four hours after transfection, control cells were treated with DMSO (Group 1) and SFB-SPOP transfected cells were treated for 24 h with DMSO (Group 2) or different drugs including mitomycin C (MMC, 1 μM) (Group 3), camptothecin (CPT, 50 nM) (Group 4), cisplatin (10 μM) (Group 5), and etoposide (10 μM) (Group 6). The cells were lysed by NETN buffer (20 mM Tris-HCl, pH 8.0, 100 mM NaCl, 1 mM EDTA, 0.5% Nonidet P-40) with 50 mM β-glycerophosphate, 10 mM NaF, and 1 μg/mL pepstatin-A at 4 °C for 3 h, followed by tandem affinity purification using streptavidin beads and S tag beads and LC-MS. A total of six groups of cell lysate were analyzed (n = 6), including one group of cells transfected with an empty vector and treated with DMSO as a negative control. All the proteins identified in this group were excluded from the list of SPOP interactome. For Geminin ubiquitination site mapping, 293 T cells were transfected with Flag-Geminin, HA-Ub, and Myc-SPOP. The co-IP sample was subjected to LC-MS (n = 1/group) without additional controls. The identified Geminin ubiquitination sites were further validated by mutagenesis and co-IP assays. For identification analysis of components of the Geminin complex, 293T cells were transfected with Flag-Geminin in the presence (Group 1) or absence (Group 2) of HA-Ub and Myc-SPOP plasmids. The co-IP samples were subjected to LC-MS (n = 1/group). Group 2 was used as a negative control. All mass spectrometry experiments were performed once.

**LC-MS analysis**. For Geminin ubiquitination mapping, the LC-MS analysis was performed using a nanoflow EASY-nLC 1000 system (Thermo Fisher Scientific, Odense, Denmark) coupled to an Orbitrap Elite mass spectrometer (Thermo Fisher Scientific, Bremen, Germany). Samples were analyzed on a home-made C18 analytical column (75 μm i.d. × 20 cm, ReproSil-Pur 120 C18-AQ, 1.9 μm, Dr. Maisch GmbH, Germany). The mobile phases consisted of Solution A (0.1% formic acid) and Solution B (0.1% formic acid in 100% ACN). The peptides were eluted using the following gradients: 5–28% of Solution B for 50 min, 28–90% of Solution B for 2 min, 90% of Solution B for 10 min, at a flow rate of 200 nL per min. The spray voltage was set at 2.0 kV and the heated capillary at 275 °C. The mass spectrometer was operated in data-dependent mode and each cycle of duty consisted of one full scan (mass range 350–1600 m/z) in the Orbitrap (60,000 resolution), followed by MS/MS experiments for 15 strongest peaks. MS/MS experiments were performed in the Orbitrap with HCD fragmentation (isolation window 1.6–15,000 resolutions; NCE 30%). Signal Required was set at 5000 and the normalized collision energy value was set at 35%. The results were processed with the UniProt human protein database (70,956 entries, download on 12-02-2016) using Protein Discoverer (Version 1.4.0.288, Thermo Fisher Scientific) and Mascot (Version 2.3.2, Matrix Science). The mass tolerances were 10 ppm for precursor and fragment Mass Tolerance 0.05 Da. The Minimum Precursor Mass was set at 350 Da and the maximum precursor mass was set at 8000 Da. Up to two missed cleavages were allowed. The search engine set the carbamidomethylation on cysteine as a fixed modification, the diglycine in lysine and acetylation on the protein N-terminal and oxidation on methionine as variable modifications. False discovery rate (FDR) thresholds cutoff for peptide were at 0.05.

For protein identification of Geminin and SPOP interactomes, the LC-MS analysis was performed using a nanoflow EASY-nLC 1200 system (Thermo Fisher Scientific, Odense, Denmark) coupled to an Orbitrap Exploris480 mass spectrometer (Thermo Fisher Scientific, Bremen, Germany). The mobile phases consisted of Solution A (0.1% formic acid) and Solution B (0.1% formic acid in 80% ACN). The peptides were eluted using the following gradients: 5–8% of Solution B for 2 min, 8–44% of Solution B for 38 min, 44–70% of Solution B for 8 min, 70–100% of Solution B for 2 min, 100% of Solution B for 10 min, at a flow rate of 200 nL per min. High-field asymmetric-waveform ion mobility spectrometry (FAIMS) was enabled during data acquisition with compensation voltages set as −45 and −65 V. MS1 data were collected in the Orbitrap (60,000 resolution). Charge states between 2 and 7 were required for MS2 analysis, and a 45-s dynamic exclusion window was used. Cycle time was set at 1.5 s. MS2 scans were performed

in the Orbitrap with HCD fragmentation (isolation window 1.6–15,000 resolutions; NCE 30%). The results were processed with the UniProt human protein database (75,004 entries, download on 07-01-2020) using Protein Discoverer (Version 2.4.1.15, Thermo Fisher Scientific) and Mascot (Version 2.7.0, Matrix Science). The mass tolerances were 10 ppm for precursor and fragment Mass Tolerance 0.05 Da. Up to two missed cleavages were allowed. The search engine set cysteine carbamidomethylation as a fixed modification and N-acetylation in the proteins and oxidation of methionine as variable modifications. False discovery rate (FDR) thresholds for protein were at 0.05 and peptide were at 0.01. The minimum peptide length was 6 and the minimum number of peptide sequences was 1.

**Karyotype analysis**. PC-3 cells were treated with DMSO or 200 nM VE-822 for 24 h and Colcemid for 1 h before harvest. Cells were washed two times in PBS, and then resuspended in 0.075 M KCl at 37 °C for 15 min. Cells were fixed with fixative (3:1 methanol:glacial acetic acid) twice, for 15 min each time. Small drops of cell suspension were placed onto a slide surface and stained with Diff-Quick staining for 1 min. Approximately 100 cells with well-spread chromosomes were photographed and analyzed and counted in each group.

**MTS dose-dependent survival assay and clonogenic survival assay**. For MTS dose-dependent survival assay, the cells were plated at a density of 1000 cells/well in 96-well plates. After 24 h, the cells were treated with different concentrations of drugs and harvested at 48 h post-treatment. The OD value was read at a wavelength of 490 nm. For clonogenic survival assay, an appropriate number of cells for different dosages of drugs were plated onto 6-well plates. After 24 h, cells were treated with DMSO or different doses of drugs. Twelve days later, colonies were fixed and stained with crystal violet 0.5% (w/v) for 1 h. The number of colonies in each group was counted and analyzed.

**Drug treatment of PDX tumors**. All mice were housed under standard pathogen-free conditions with a 12 h light/dark cycle and access to food and water ad libitum. We have complied with all relevant ethical regulations for animal care and use, and the animal studies received ethical approval by the Institutional Animal Care and Use Committee (IACUC) at the Mayo Clinic. The PDX tumors including SPOP-WT and Q165P mutant[74] were expanded by passaging tumor pieces (~1 mm$^3$) subcutaneously into 6-to-8-week-old SCID male mice. After tumors reach ~100 mm$^3$ in size (~4 weeks after transplantation), tumor-positive animals in both SPOP-WT and Q165P groups were randomly divided into different treatment groups (5 mice/group). Mice were treated with vehicle control or 60 mg/kg/day VE-822 five days a week for 21 consecutive days. Tumor growth was measured by caliper every 3 to 4 days. The tumor volume was calculated using the formula $0.5 \times$ length $(L) \times$ width $(W)^2$. When the first tumor reached a volume of 1000 mm$^3$, the treatment was terminated and tumors were harvested for photography.

**Generation of graphs and statistical analysis**. Graphs were generated using GraphPad Prism 8 project (GraphPad, Inc.) or Microsoft Office Excel 2010. All numerical data are presented as mean ± SEM or mean ± SD as required. Differences between groups were compared by $t$ tests, two-way ANOVA, or Wilcoxon rank-sum test with continuity correction by GraphPad Prism 8 project for Statistical Computing. The following symbols were used to denote statistical significance: $*P < 0.05$, $**P < 0.01$, $***P < 0.001$, ns, not significant.

**Reporting summary**. Further information on research design is available in the Nature Research Reporting Summary linked to this article.

## Data availability

The raw data for western blot generated in this study have been deposited in the Mendeley database (https://doi.org/10.17632/8n7xt5rkhc.1). The atomic coordinates for the protein structure presented in this publication are deposited in the Protein Data Bank under accession code 7KLZ. Already published protein structures used in this study can be found in the Protein Data Bank under accession codes: 2WVR and 6WGG. The mass spectrometry proteomics raw data have been deposited to the ProteomeXchange Consortium (http://proteomecentral.proteomexchange.org) via the iProX partner repository[75] with the dataset identifier PXD027915 and PXD028210. All relevant data are available from the authors. Source data are provided with this paper.

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

## Acknowledgements

This work was supported in part by the Mayo Clinic Foundation (to H.H.), Mayo Clinic Center for Biomedical Discovery (grant #098 to G.M. and H.H.), the National Institutes of Health (R01 CA132878 to G.M.), the National Natural Science Foundation of China (81672558, 81201533, and 81972396 to C.W., 81672544 and 81872099 to D.Y.). X-ray diffraction data were collected at beamline 19-BM of Argonne National Laboratory, Structural Biology Center at the Advanced Photon Source. The Structural Biology Center is operated by UChicago Argonne LLC, for the U.S. Department of Energy, Office of Biological and Environmental Research, under contract DE-AC02-06CH11357. We are very grateful to Jerzy Osipiuk at the Structural Biology Center for assistance with X-ray data collection and Lin Huang from Fudan University for assistance with LC-MS analysis.

## Author contributions

H.H. conceived the study. J.M., Q.S., G.C., H.S. and M.V.B. performed experiments, data collection, and analysis. Y.W. provided the SPOP Q165P PDX model and Y.Z., Y.Y., and Y.H. expanded PDX tumors for drug treatment. L.W. performed bioinformatics analysis. H.H., G.M., C.W., and D.Y. wrote the manuscript.

## Competing interests

The authors declare no competing interests.
