## [Peer Review File · Nature Communications]

SPOP mutation induces replication over-firing by impairing Geminin ubiquitination and triggers replication catastrophe upon ATR inhibitionREVIEWER COMMENTS

Reviewer #1 (Remarks to the Author):

SPOP mutation induces replication-overfiring by impairing the regulatory function of Geminin-Cdt1.
Comments:

In this manuscript, the authors have investigated the mechanism of genome instability underlying prostate cancers caused by SPOP mutations. The authors demonstrate that non-degradative poly-ubiquitylation of Geminin by SPOP at endogenous level regulates the association of Cdt1 with MCM helicase complex to trigger replication origin firing. Prostate cancer associated mutations in SPOP lead to deregulation of Geminin-Cdt1-MCM complex thereby inducing re-replication and increased chromosomal instability in prostate cancer cells. This is an exhaustive study important to the field of genome stability research since it presents an important therapeutic opportunity of intervening SPOP mutation induced prostate cancers by ATR inhibition. Nonetheless, there are few layers of missing information which if addressed would significantly strengthen the manuscript.

Major concerns:

1. Authors demonstrate that poly-ubiquitylation of Geminin by SPOP is essential for inhibiting re-replication and chromosomal instability. Therefore, it is very surprising how ubiquitylation defective mutants of SPOP or SPOP CRISPR knockouts even survive in culture. SPOP has been shown to cause neonatal lethality in mouse and is therefore essential in development. Hence, it would be important to know if the authors observed any differences in the growth characteristics of SPOP mutant/null human cells in culture to prove that poly-ubiquitylation of Geminin is truly essential for replication and cell growth.
2. In the first part of the manuscript authors show that SPOP mutants that cannot interact with Geminin fail to stimulate its poly-ubiquitylation when overexpressed in cells (e.g. Fig 2A) – when they are overexpressed a basal level of geminin ubiquitylation is observed, likely delivered by endogenous SPOP. In the latter part of the manuscript the cancer derived mutants of SPOP that cannot ubiquitylate geminin are however used as dominant negative mutants and when they are expressed the ubiquitylation of geminin is abolished. Can authors explain this discrepancy, please? What is the underlying reason for the dominant negative effect?
3. Western blots demonstrating that SPOP mutations reduce Geminin poly-ubiquitylation thereby leading to reduced interactions between Cdt1 and MCMs (Fig. 3 and 4) should be supported with quantification (and error range) from multiple experiments for reproducibility and reliability.
4. It is not clear why authors observe no MCM2 on the chromatin during the G1 phase (Fig 5B) since replication licensing occurs during late mitosis and G1 and Mcm2-7 levels are expected to be at their peak during G1 phase and then be unloaded from chromatin as replication progresses. This is important as Mcm2 is the only Mcm2-7 subunit tested and its level on chromatin underpins the main point of the manuscript. Also, the levels of SPOP and Geminin depicted in Fig 5D do not appear to correlate with western blot image in Fig. 5B. Fig.5D should be presented as a compilation from multiple experiments with error range.

Minor concerns:

1. Although the authors mention that Geminin was chosen for this study as the only interactor that functions both in DNA replication and DNA repair, the DNA repair aspects of Geminin function have not been explored in context of its poly-ubiquitylation dependent regulation by SPOP. Could authors comment on them in the discussion please.
2. Since VSSST motif in Geminin is shown to be crucial for its poly-ubiquitylation by SPOP, it would be useful to know if there are cancers associated with mutations in this motif of Geminin or patients with K100/127 mutations. This could complement well the result showing prostate cancer SPOP mutations fail to ubiquitylate Geminin. Similarly, it would be worth discussing why SPOP mutations are over-represented in prostate cancers despite having a general role in replication licensing.
3. The asterisks marked as non-specific bands in Suppl. Fig. 3A are not visible in all lanes – why?
4. The finding in Figure 3C and top of page 9 that SPOP Δ BTB fails to ubiquitylate Geminin was already shown in Fig. 2A.
5. It would be beneficial for the manuscript to explain what is the rationale for selecting F102C and

F133V SPOP mutants over the remaining prostate cancer mutants that fail to ubiquitylate and do not interact with Geminin (Fig. 3D) for the remaining of the experiments.

6. Cdt1 binding to Cdc6-ORC is upstream to MCM binding for replication origin firing. It would be worth exploring if SPOP dependent poly-ubiquitylation of Geminin interferes with Cdt1 binding to Cdc6-ORC at all.

7. It is not clear why K27-only HA-Ub was used in this experiment in Fig. 5C instead of WT-Ub. It would be important to know if any inconsistencies were observed while using WT-Ub.

8. The data point indicators for SPOP and Geminin have been possibly swapped Fig. 5D.

9. Suppl. Fig. 2F is not clearly mentioned in the text.

10. The rationale for using Hydroxyurea in experiment in Supp. Fig. 2I is not explained and confusing.

11. In contrast to the author's claim, the PONDR signals for human and yeast Cdt1 appear totally different throughout the length of the protein (Suppl. Fig. 3D and E).

12. It would be helpful if the representative images to show inter-origin distances in Suppl. Fig. 4C and D included at least two consecutive origins (red stretches).

Reviewer #2 (Remarks to the Author):

The role of geminin in regulating replication origins firing through Cdt1 is well known but the mechanism of geminin's regulatory role is unknown.

This is a well-executed paper that fully supports the objectives and conclusions stated in the abstract. The authors adequately demonstrated that SPOP promotes K27-linked non-degradative poly-ubiquitination of Geminin at lysine residues 100 and 127 which indirectly blocks the association of Cdt1 with the MCM protein complex. They also showed that cancer-associated SPOP mutations impair Geminin K27-linked poly-ubiquitination and induce replication origin over-firing and re-replication and that SPOP-mutated tumors may be susceptible to ATR inhibitor therapy.

My major criticism concerns the mechanistic model presented to explain their findings. Two assumptions were made in their model presented in figure 4c.

- First, the structure of hCdt1 is based on the yeast Cdt1 structure bound to the γ MCM complex though the primary sequences of the human and yeast Cdt1 are very diverged and structural predictions are very different (Suppl Fig 3d, e). The authors took the liberty of the divergence and structural disorder of the C-termini to build what they called a "plausible" structural model. Therefore, the modelling is not evidence based.
- Second, in the steric hindrance model (fig 4c and 7c), hCdt1, geminin and MCM form a complex that is not supported by biochemical or physical evidence. Missing is IP of geminin to pull down the MCM complex.

In light of these concerns, alternative models consistent with the interference of hCdt1-MCM complex formation by geminin ubiquitination must be entertained.

Minor concerns:

- Page 2. "During late M and early G1 phases of the cell cycle, Cdc6 and Cdt1 are loaded onto replication origins in an ORC-dependent manner and subsequently recruit MCM proteins to the origins." What is the evidence for this sequence of events? References? In yeast, Cdt1 is loaded onto replication origins in a complex with MCM in G1 phase in an ORC dependent manner, not separately and not in late M phase.

On the whole, this study is informative for cancer biology especially the role of SPOP in regulating replication origin firing through ubiquitination of geminin but makes less of an impact towards understanding the mechanism for geminin in regulating Cdt1 in replication origin firing.

Reviewer #3 (Remarks to the Author):

Geminin binds to Cdt1 and modulates helicase loading, preventing re-replication. The authors show that SPOP, an E3 ubiquitin ligase adaptor, interacts with Geminin and regulates replication. The interaction between the MATH domain of SPOP is characterized using structural tools that include NMR and crystallography, and molecular modelling to inform the Cdt1-MCM interaction. The authors find that SPOP promotes Geminin poly-ubiquitination, which prevents re-replication by blocking the MCM-Cdt1 interaction. Cancer-causing mutations in SPOP impair Geminin ubiquitination, hence inducing re-replication. Replication stress triggers replication catastrophe and ATR inhibition that causes cell death. The replication stress caused by SPOP mutations triggers replication catastrophe and cell death upon ATR inhibition.

This is a solid study, which deserves to be published in Nature Communications, after some issues are addressed.

1. Page 10. Ctd1 should be Cdt1 instead.
2. The SPOP MATH domain was co-crystallised with a 15-mer Geminin peptide, but only 9 residues were built in the atomic model. How confident are the authors about the sequence register and the peptide binding polarity? If the authors assigned these two by being guided by other MATH domain co-crystal structures and by the NMR experiments, then it would be appropriate to clarify that in the Methods section. If instead the polarity has been assigned based on the data, then it would be important to improve the quality of the relevant panels in Supplementary Figure 1 to make this more evident. On a related note, panel m does not have a caption.
3. The authors incorrectly refer to 5V8F as Cdt1-ORC-MCM complex. It is indeed an ORC-Cdc6-Cdt1-MCM complex (generally referred to as OCCM), which is a short-lived helicase loading intermediate, which can be stabilised in the presence of an ATP analogue. The question arises as to why the authors have chosen this complex (whose existence has yet to be demonstrated for human proteins), rather than the yeast MCM-Cdt1 complex, which contains an open MCM ring and a slightly different interaction interface with Cdt1. Although an interaction between MCM and Cdt1 has been reported between human proteins, it should be noted that protein co-expression and in vitro reconstitution work with higher eukaryotic proteins so far has failed to show stable complex formation for MCM-Cdt1. Therefore, the OCCM and MCM-Cdt1 structures as described in recent cryo-EM work might very well be yeast-specific. These facts should be acknowledged when describing the molecular modelling.
4. Given the caveats mentioned in point 3, I believe that a structural model obtained by combining information on yeast structures with MD simulations of Geminin-Cdt1 complex should not be referred to as a "compelling explanation" for the experimental results (page 11). Rather, they should be referred to as "suggestive".

Authors' Response to Reviewers' Comments on Manuscript NCOMMS-21-03998

We very much thank the reviewers for the positivity and insightful comments for the improvement of our manuscript.

Reviewer #1 (Remarks to the Author):

SPOP mutation induces replication-overfiring by impairing the regulatory function of Geminin-Cdt1.

Comments:

In this manuscript, the authors have investigated the mechanism of genome instability underlying prostate cancers caused by SPOP mutations. The authors demonstrate that non-degradative poly-ubiquitylation of Geminin by SPOP at endogenous level regulates the association of Cdt1 with MCM helicase complex to trigger replication origin firing. Prostate cancer associated mutations in SPOP lead to deregulation of Geminin-Cdt1-MCM complex thereby inducing re-replication and increased chromosomal instability in prostate cancer cells. This is an exhaustive study important to the field of genome stability research since it presents an important therapeutic opportunity of intervening SPOP mutation induced prostate cancers by ATR inhibition. Nonetheless, there are few layers of missing information which if addressed would significantly strengthen the manuscript.

Response: We thank the reviewer for recognizing the novelty and significance of our finding. We also thank the reviewer for the insightful suggestions that have helped us improve our manuscript significantly.

Major concerns:

1. Authors demonstrate that poly-ubiquitylation of Geminin by SPOP is essential for inhibiting re-replication and chromosomal instability. Therefore, it is very surprising how ubiquitylation defective mutants of SPOP or SPOP CRISPR knockouts even survive in culture. SPOP has been shown to cause neonatal lethality in mouse and is therefore essential in development. Hence, it would be important to know if the authors observed any differences in the growth characteristics of SPOP mutant/null human cells in culture to prove that poly-ubiquitylation of Geminin is truly essential for replication and cell growth.

Response: We thank the reviewer for raising these great points. As the reviewer mentioned, conventional homozygous deletion of *Spop* gene has been shown to cause neonatal lethality in mouse and is therefore essential in development (Cai and Liu, Proc Natl Acad Sci U S A, 2016, Dev Biol, 2017; Claiborn et al., J Clin Invest, 2010; Yin et al., Dev Cell, 2019). A possible explanation for this observation is that SPOP deficiency may lead to cellular lethality in certain cell types in the body, which is consistent with our findings that inactivation of SPOP due to gene mutation leads to replication origin overfiring and genomic instability and that this deleterious effect can be more drastic if the expression level of endogenous ATR is low in those cell types or when ATR is pharmacologically inhibited regardless of the cell origin.

Notably, SPOP mutations are only frequently detected in very few cancer types such as prostate and endometrial cancers, but less frequently in other cancer types (Clark and Burleson, Am J

Cancer Res, 2020; Wang et al., Nat Rev Urol, 2020), suggesting that SPOP mutations may also lead to the activation of other survival pathways in prostatic and endometrial tissues that can counteract the deleterious effect caused by SPOP inactivation (mutation or deletion). In support of this notion, research from different groups has invariably shown previously that SPOP mutation or deletion promotes growth of prostate cancer cells by stabilizing a few key survival factors in prostate cancer such as androgen receptor (AR) and its coactivators such as SRC-3 and TRIM24 (Blattner et al., Cancer Cell, 2017; Dai et al., Nat Med, 2017; Geng et al., Proc Natl Acad Sci U S A, 2013; Groner et al., Cancer Cell, 2016).

Review Fig. 1. PC-3 cells, infected with lentivirus expressing control or Geminin-specific shRNA in combination with empty vector (EV), shRNA-resistant Geminin WT or K100/127R mutant, were transfected with empty vector (EV) or Myc-SPOP-F133V plasmid. Cells were harvested for Western blot analysis (a) and colony formation assay. Representative colonies are shown in (b) with quantification data shown in (c). Data are presented as the mean ± SD of three independent experiments. Two-tailed unpaired Student's *t*-test; * *P* < 0.05, *** *P* < 0.001, ns, not significant.

To experimentally address the reviewer's concern, as suggested, we examined the role of Geminin in regulating prostate cancer cell growth in the presence or absence of SPOP mutation using colony formation assay. We demonstrated that knocking down Geminin significantly decreased PC-3 prostate cancer cell growth and that this effect was rescued by restored expression of shRNA-resistant WT Geminin but not the SPOP ubiquitination-resistant mutant (K100/127R) (Review Figure. 1). However, expression of SPOP-F133V mutant abolished Geminin knockdown-induced inhibition of cell growth (Review Fig. 1). These new data suggest that poly-ubiquitylation of Geminin by SPOP is essential for cell growth. However, when SPOP

is mutated, other survival pathways activated by SPOP mutation (e.g. AR) may surpass the detrimental effect of impaired poly-ubiquitination of Geminin on prostate cancer cell growth.

We think these findings are quite exciting as SPOP mutation-induced impairment of Geminin poly-ubiquitination, replication over-firing and the associated detrimental effect on prostate cancer cell growth represent a unique therapeutic vulnerability for the treatment of SPOP mutated cancer. Because of this consideration and the last point raised by another reviewer (Reviewer #2), we changed the title of our manuscript to “SPOP mutation induces replication over-firing by impairing Geminin ubiquitination and triggers replication catastrophe upon ATR inhibition” by more emphasizing the cancer biology aspect of the impact of SPOP mutation on Geminin ubiquitination and replication origin firing.

2. In the first part of the manuscript authors show that SPOP mutants that cannot interact with Geminin fail to stimulate its poly-ubiquitylation when overexpressed in cells (e.g. Fig 2A) – when they are overexpressed a basal level of geminin ubiquitylation is observed, likely delivered by endogenous SPOP. In the latter part of the manuscript the cancer derived mutants of SPOP that cannot ubiquitylate geminin are however used as dominant negative mutants and when they are expressed the ubiquitylation of geminin is abolished. Can authors explain this discrepancy, please? What is the underlying reason for the dominant negative effect?

Response: This is an excellent point. There are several findings supporting the dominant negative effect of SPOP mutations. **First**, almost all SPOP mutations detected thus far (except one mutation) in prostate cancer patients are hemizygous mutations (Armenia et al., Nat Genet, 2018; Barbieri et al., Nat Genet, 2012; Cancer Genome Atlas Research, Cell, 2015). **Second**, increasing evidence indicates that prostate cancer-derived SPOP mutants functionally behave in a dominant negative manner (Blattner et al., Cancer Cell, 2017; Dai et al., Nat Med, 2017; Zhang et al., Nat Med, 2017). **Third**, this is consistent with the findings from biochemical and structural studies showing that the SPOP protein can form a dimer or oligomer via its BTB domain and BACK domain (Marzahn et al., EMBO J, 2016; Zhuang et al., Mol Cell, 2009), providing a molecular basis for SPOP mutants to function in a dominant negative fashion. Our finding confirmed the dominant-negative effect of SPOP mutations. Almost all mutants lost the ability to bind and ubiquitinate Geminin even though endogenous SPOP is present (**Fig. 3d and 3e**).

3. Western blots demonstrating that SPOP mutations reduce Geminin poly-ubiquitylation thereby leading to reduced interactions between Cdt1 and MCMs (Fig. 3 and 4) should be supported with quantification (and error range) from multiple experiments for reproducibility and reliability.

Response: We thank the reviewer for these helpful suggestions. We repeated the Cdt1 and MCM protein binding assays three times. The WB data are shown in **Fig. 4a and 4b** and the quantification results are shown in **Supplementary Fig. 3d and 3e**.

4. It is not clear why authors observe no MCM2 on the chromatin during the G1 phase (Fig 5B) since replication licensing occurs during late mitosis and G1 and Mcm2-7 levels are expected to be at their peak during G1 phase and then be unloaded from chromatin as replication progresses. This is important as Mcm2 is the only Mcm2-7 subunit tested and its level on chromatin underpins the main point of the manuscript. Also, the levels of SPOP and Geminin depicted in

Fig 5D do not appear to correlate with western blot image in Fig. 5B. Fig.5D should be presented as a compilation from multiple experiments with error range.

Response: We thank the reviewer for raising these excellent points. We agree that MCM2 loading on chromatin starts at early G1 phase and increases as cells progress through G1 and reaches a peak at late G1 phase (Kuipers et al., J Cell Biol, 2011; Matson et al., Elife, 2017; Mukherjee et al., PLoS One, 2009; Tanaka et al., Cell, 1997). As suggested by the reviewer, we repeated the Western blot for **Fig. 5b** three times, and the quantification results are shown in **Fig. 5c**. These data indicate that MCM2 was readily detectable at early G1 (at 0 h after released from the thymidine and L-mimosine double block (G1 arrest) and peaked at 2 h after released from thymidine and L-mimosine double block.

Minor concerns:

1. Although the authors mention that Geminin was chosen for this study as the only interactor that functions both in DNA replication and DNA repair, the DNA repair aspects of Geminin function have not been explored in context of its poly-ubiquitylation dependent regulation by SPOP. Could authors comment on them in the discussion please.

Response: We thank the reviewer for raising these excellent points. We performed new analysis of our data and demonstrated that Geminin is the only one gene that was overlapped between the list of genes involved in DNA replication and SPOP-interacting proteins identified by Y2H and mass spectrometry (**Fig. 1c**); however, there is no overlap between the lists of genes involved in DNA repair and SPOP-interacting proteins identified by Y2H and mass spectrometry (**Supplementary Fig. 1i**). This is the major reason for us to pursue the role of SPOP regulation of Geminin in DNA replication. The discussion of this rationale is provided in the revised manuscript.

2. Since VSSST motif in Geminin is shown to be crucial for its poly-ubiquitylation by SPOP, it would be useful to know if there are cancers associated with mutations in this motif of Geminin or patients with K100/127 mutations. This could complement well the result showing prostate cancer SPOP mutations fail to ubiquitylate Geminin. Similarly, it would be worth discussing why SPOP mutations are over-represented in prostate cancers despite having a general role in replication licensing.

Response: This is an excellent point. We checked the TCGA dataset and found that there is no mutation in the Geminin ubiquitination sites. However, there are several Geminin mutation sites located in the SPOP binding consensus (SBC) motif ¹⁹⁹VSSST²⁰³, including S202F, V199 nonsense mutation and T203 nonsense mutation mutations (Cancer Genome Atlas, Nature, 2012). We further experimentally demonstrated that S202F mutant of Geminin cannot be bound or ubiquitinated by SPOP (**Fig. 3g and 3h**).

3. The asterisks marked as non-specific bands in Suppl. Fig. 3A are not visible in all lanes – why?

Response: The non-specific bands may be caused by longer exposure of WB films. We repeated this assay and new results are shown in revised **Supplementary Fig. 3a**.

4. The finding in Figure 3C and top of page 9 that SPOP Δ BTB fails to ubiquitylate Geminin was already shown in Fig. 2A.

Response: We agree with the reviewer. We repeated the ubiquitination assay for **Fig. 2a** by excluding the SPOP Δ BTB mutant in the new experiments.

5. It would be beneficial for the manuscript to explain what is the rationale for selecting F102C and F133V SPOP mutants over the remaining prostate cancer mutants that fail to ubiquitylate and do not interact with Geminin (Fig. 3D) for the remaining of the experiments.

Response: As suggested by the reviewer, we have indicated in the manuscript that F102C and F133V are the two missense mutations in SPOP that are most frequently detected in prostate cancer patient samples.

6. Cdt1 binding to Cdc6-ORC is upstream to MCM binding for replication origin firing. It would be worth exploring if SPOP dependent poly-ubiquitylation of Geminin interferes with Cdt1 binding to Cdc6-ORC at all.

Response: We performed the experiments as suggested by the reviewer. We demonstrated that SPOP-dependent poly-ubiquitination of Geminin is also important for Cdt1 binding with Cdc6 and ORC2 (**Fig. 4a and 4b**), consistent with the results of our structure modeling and of the reports from other groups that Cdt1-MCM binding is essential for the latching between the MCM-Cdt1 and ORC-Cdc6 (Guerrero-Puigdevall et al., Nat Commun, 2021; Yuan et al., Proc Natl Acad Sci U S A, 2020).

7. It is not clear why K27-only HA-Ub was used in this experiment in Fig. 5C instead of WT-Ub. It would be important to know if any inconsistencies were observed while using WT-Ub.

Response: We repeated the IP assay using WT-Ub. We demonstrated that Geminin was ubiquitinated at almost all phases of the cell cycle, but its ubiquitination was increased in G1 and S phases. In contrast, in SPOP mutated cells Geminin ubiquitination was decreased in S and early G2 phases (**Supplementary Fig. 4b**). This observation is consistent with the result of the experiment using K27-only HA-Ub. It is reported that Geminin is ubiquitinated and degraded at G1 phase by the anaphase-promoting complex (APC) through K48-linked poly-ubiquitination (McGarry and Kirschner, Cell, 1998), which provides a plausible explanation as to why Geminin ubiquitination is high in G1 phase when WT-Ub is used in the assay.

8. The data point indicators for SPOP and Geminin have been possibly swapped Fig. 5D.

Response: We thank the reviewer for raising these excellent points. Since the main purpose of these experiments is to examine the effect of SPOP ubiquitination of Geminin on MCM2 loading on chromatin, we focused on the MCM2 chromatin loading in different cell cycle phases in the control and SPOP F133V mutant cells. To this end, we repeated WB analysis for **Fig. 5b** three times and the quantified data of MCM2 chromatin loading are shown in **Fig. 5c**.

9. Suppl. Fig. 2F is not clearly mentioned in the text.

Response: We have mentioned this figure in the text of the revised manuscript.

10. The rationale for using Hydroxyurea in experiment in Suppl. Fig. 2I is not explained and confusing.

Response: Since hydroxyurea (HU) is a replication inhibitory agent, we used it to assess whether SPOP mutation affects Geminin protein level in the context of inhibition of DNA replication. We have added the rationale for this experiment in the text.

11. In contrast to the author's claim, the PONDR signals for human and yeast Cdt1 appear totally different throughout the length of the protein (Suppl. Fig. 3D and E).

Response: We agree that the PONDR predictions for the human and yeast Cdt1 appear totally different. The PONDR predictions could not be fairly compared and doing so does not bring any new or meaningful information. In the updated **Supplementary Fig. 3f**, we instead compared the conserved M-WHD regions in human and yeast Cdt1, the corresponding X-ray and cryo-EM structures of which are available. We showed that in yeast and human, the M-WHDs are very similar structurally, providing the basis for docking the M-WHD-containing and Geminin-bound human Cdt1 structure to the corresponding M-WHD of yeast Cdt1 in the yCdt1-OCCM complex structure.

12. It would be helpful if the representative images to show inter-origin distances in Suppl. Fig. 4C and D included at least two consecutive origins (red stretches).

Response: As suggested by the reviewer, we have used new representative images including at least two consecutive origins (red stretches) in revised **Supplementary Fig. 4d and 4e**.

Reviewer #2 (Remarks to the Author):

The role of geminin in regulating replication origins firing through Cdt1 is well known but the mechanism of geminin's regulatory role is unknown.

This is a well-executed paper that fully supports the objectives and conclusions stated in the abstract. The authors adequately demonstrated that SPOP promotes K27-linked non-degradative poly-ubiquitination of Geminin at lysine residues 100 and 127 which indirectly blocks the association of Cdt1 with the MCM protein complex. They also showed that cancer-associated SPOP mutations impair Geminin K27-linked poly-ubiquitination and induce replication origin over-firing and re-replication and that SPOP-mutated tumors may be susceptible to ATR inhibitor therapy.

Response: We are very thankful for the reviewer's positivity and enthusiasm about our study.

My major criticism concerns the mechanistic model presented to explain their findings. Two assumptions were made in their model presented in figure 4c.

- First, the structure of hCdt1 is based on the yeast Cdt1 structure bound to the yMCM complex though the primary sequences of the human and yeast Cdt1 are very diverged and structural predictions are very different (Suppl Fig 3d, e). The authors took the liberty of the divergence and structural disorder of the C-termini to build what they called a “plausible” structural model. Therefore, the modelling is not evidence based.

Response: The reviewer is correct. Our structural model was plausible but not evidence-based. We have now performed experiments to evaluate our model. As demonstrated by Geminin-MCM co-IP assay and mass spectrometry (**Supplementary Fig. 3i-k**), Geminin could pull down MCM proteins in cultured cells. In addition, the abundance of MCM proteins pulled down by Geminin was largely reduced in SPOP-transfected cells compared to control cells (**Supplementary Fig. 3i-k**). Notably, no interaction between Geminin and MCM protein was detected under *in vitro* conditions (**Supplementary Fig. 3l**). Taken together, these results support our model (**Fig. 7**) showing that there is no direct interaction between Geminin and MCM and that their interaction is likely bridged through other proteins such as Cdt1.

- Second, in the steric hindrance model (fig 4c and 7c), hCdt1, geminin and MCM form a complex that is not supported by biochemical or physical evidence. Missing is IP of geminin to pull down the MCM complex.

Response: We thank the reviewer for raising these excellent points. As we discussed above, our results demonstrated that Geminin could pull down MCM proteins *in vivo* and that the abundance of MCM proteins pulled down by Geminin was largely reduced in SPOP-transfected cells compared to control cells (**Supplementary Fig. 3i-k**). However, no interaction between Geminin and MCM protein was detected under *in vitro* conditions (**Supplementary Fig. 3l**), suggesting that Geminin-MCM interaction is likely bridged through other proteins such as Cdt1.

In light of these concerns, alternative models consistent with the interference of hCdt1-MCM complex formation by geminin ubiquitination must be entertained.

Response: We thank the reviewer for raising these excellent points. Our new data provide further support of our model that Geminin indirectly interacts with MCM proteins (**Supplementary Fig. 3i-l**). Furthermore, we showed that their interaction was largely reduced in SPOP-transfected cells compared to control cells (**Supplementary Fig. 3i-k**). This effect could be mediated through the physical barrier caused by SPOP poly-ubiquitination of Geminin as we predicted in **Fig. 4c**. As now stated in the revised manuscript, we acknowledge that other explanations are possible, but that our model provides the simplest possible explanation and is consistent with experimental results. A general conformation change in Geminin-Cdt1-MCM complex caused by SPOP-induced poly-ubiquitination of Geminin is possible. We have modified our model shown in **Fig. 7b** and discussed this alternative possibility in the Discussion section.

Minor concerns:

- Page 2. “During late M and early G1 phases of the cell cycle, Cdc6 and Cdt1 are loaded onto replication origins in an ORC-dependent manner and subsequently recruit MCM proteins to the

origins.” What is the evidence for this sequence of events? References? In yeast, Cdt1 is loaded onto replication origins in a complex with MCM in G1 phase in an ORC dependent manner, not separately and not in late M phase.

Response: We thank the reviewer for mentioning these points. We are sorry for the inaccurate description. We have revised the wording and added the references in the text: ORC binds origins of replication and recruits Cdc6 at the M/G1 transition. Cdc6-bound ORC recruits Mcm2-7 in complex with Cdt1 to origin of DNA replication (Nguyen et al., Nature, 2001; Ticau et al., Cell, 2015; Yardimci and Walter, Nat Struct Mol Biol, 2014).

On the whole, this study is informative for cancer biology especially the role of SPOP in regulating replication origin firing through ubiquitination of geminin but makes less of an impact towards understanding the mechanism for geminin in regulating Cdt1 in replication origin firing.

Response: We agree with the reviewer. To emphasize the cancer biology aspect of the impact of SPOP mutation on Geminin ubiquitination and replication origin firing, but less emphasize the impact towards the mechanistic insight into regulating the function of Cdt1-Geminin in replication origin firing, we have changed the title of our manuscript to “SPOP mutation induces replication over-firing by impairing Geminin ubiquitination and triggers replication catastrophe upon ATR inhibition”.

Reviewer #3 (Remarks to the Author):

Geminin binds to Cdt1 and modulates helicase loading, preventing re-replication. The authors show that SPOP, an E3 ubiquitin ligase adaptor, interacts with Geminin and regulates replication. The interaction between the MATH domain of SPOP is characterized using structural tools that include NMR and crystallography, and molecular modelling to inform the Cdt1-MCM interaction.

The authors find that SPOP promotes Geminin poly-ubiquitination, which prevents re-replication by blocking the MCM-Cdt1 interaction. Cancer-causing mutations in SPOP impair Geminin ubiquitination, hence inducing re-replication. Replication stress triggers replication catastrophe and ATR inhibition that causes cell death. The replication stress caused by SPOP mutations triggers replication catastrophe and cell death upon ATR inhibition.

This is a solid study, which deserves to be published in Nature Communications, after some issues are addressed.

Response: We very much appreciate the reviewer for the positivity and enthusiasm about our study.

1. Page 10. Ctd1 should be Cdt1 instead.

Response: We are sorry for the typo. It has been corrected.

2. The SPOP MATH domain was co-crystallised with a 15-mer Geminin peptide, but only 9

residues were built in the atomic model. How confident are the authors about the sequence register and the peptide binding polarity? If the authors assigned these two by being guided by other MATH domain co-crystal structures and by the NMR experiments, then it would be appropriate to clarify that in the Methods section. If instead the polarity has been assigned based on the data, then it would be important to improve the quality of the relevant panels in Supplementary Figure 1 to make this more evident. On a related note, panel m does not have a caption.

Response: We determined the polarity of the Geminin SBC peptide we co-crystallized with SPOP MATH based on published high-resolution structures of SPOP MATH-SBC. We have now included in the Methods section a description of how we obtained the polarity of the Geminin peptide, and we also added one reference. The SBC motif is highly conserved, sequence- and structure-wise, with the non-polar residue in the motif surrounded by aromatic residues from SPOP. **Fig. 1f** shows the highly conserved SBC motifs from various protein sources including Geminin from different species. We further showed in our crystal structure that this non-polar residue (Val199) is recognized by SPOP MATH F102, Y123, W131 and F133 aromatic residues (**Fig. 1k**).

3. The authors incorrectly refer to 5V8F as Cdt1-ORC-MCM complex. It is indeed an ORC-Cdc6-Cdt1-MCM complex (generally referred to as OCCM), which is a short-lived helicase loading intermediate, which can be stabilised in the presence of an ATP analogue. The question arises as to why the authors have chosen this complex (whose existence has yet to be demonstrated for human proteins), rather than the yeast MCM-Cdt1 complex, which contains an open MCM ring and a slightly different interaction interface with Cdt1. Although an interaction between MCM and Cdt1 has been reported between human proteins, it should be noted that protein co-expression and in vitro reconstitution work with higher eukaryotic proteins so far has failed to show stable complex formation for MCM-Cdt1. Therefore, the OCCM and MCM-Cdt1 structures as described in recent cryo-EM work might very well be yeast-specific. These facts should be acknowledged when describing the molecular modelling.

Response: We thank the reviewer for the excellent comments. We have corrected this error in the revised manuscript and mentioned the limits of our model that uses a yeast OCCM template. We had used 5V8F as a model template because of the relatively high resolution and inclusion of the Cdt1 C-WHD in the structure. We agree that the cryo-EM structure of OCCM (PDB code 6WGG) is better since it reflects the intermediate state of OCCM formation. We have performed new modeling using the OCCM cryo-EM structure (PDB code 6WGG) as a template.

4. Given the caveats mentioned in point 3, I believe that a structural model obtained by combining information on yeast structures with MD simulations of Geminin-Cdt1 complex should not be referred to as a “compelling explanation” for the experimental results (page 11). Rather, they should be referred to as “suggestive”.

Response: We agree with the reviewer’s comments. We have now rewritten the text (“The model provides a plausible explanation...”) to better convey the suggestive nature of our proposed model.

References

- Armenia, J., Wankowicz, S.A.M., Liu, D., Gao, J., Kundra, R., Reznik, E., Chatila, W.K., Chakravarty, D., Han, G.C., Coleman, I., *et al.* (2018). The long tail of oncogenic drivers in prostate cancer. *Nat Genet* **50**, 645-651.
- Barbieri, C.E., Baca, S.C., Lawrence, M.S., Demichelis, F., Blattner, M., Theurillat, J.P., White, T.A., Stojanov, P., Van Allen, E., Stransky, N., *et al.* (2012). Exome sequencing identifies recurrent SPOP, FOXA1 and MED12 mutations in prostate cancer. *Nat Genet* **44**, 685-689.
- Blattner, M., Liu, D., Robinson, B.D., Huang, D., Poliakov, A., Gao, D., Nataraj, S., Deonaraine, L.D., Augello, M.A., Sailer, V., *et al.* (2017). SPOP Mutation Drives Prostate Tumorigenesis In Vivo through Coordinate Regulation of PI3K/mTOR and AR Signaling. *Cancer Cell* **31**, 436-451.
- Cai, H., and Liu, A. (2016). Spop promotes skeletal development and homeostasis by positively regulating Ihh signaling. *Proc Natl Acad Sci U S A* **113**, 14751-14756.
- Cai, H., and Liu, A. (2017). Spop regulates Gli3 activity and Shh signaling in dorsoventral patterning of the mouse spinal cord. *Dev Biol* **432**, 72-85.
- Cancer Genome Atlas, N. (2012). Comprehensive molecular characterization of human colon and rectal cancer. *Nature* **487**, 330-337.
- Cancer Genome Atlas Research, N. (2015). The Molecular Taxonomy of Primary Prostate Cancer. *Cell* **163**, 1011-1025.
- Claiborn, K.C., Sachdeva, M.M., Cannon, C.E., Groff, D.N., Singer, J.D., and Stoffers, D.A. (2010). Pcf1 modulates Pdx1 protein stability and pancreatic beta cell function and survival in mice. *J Clin Invest* **120**, 3713-3721.
- Clark, A., and Burleson, M. (2020). SPOP and cancer: a systematic review. *Am J Cancer Res* **10**, 704-726.
- Dai, X., Gan, W., Li, X., Wang, S., Zhang, W., Huang, L., Liu, S., Zhong, Q., Guo, J., Zhang, J., *et al.* (2017). Prostate cancer-associated SPOP mutations confer resistance to BET inhibitors through stabilization of BRD4. *Nat Med* **23**, 1063-1071.
- Geng, C., He, B., Xu, L., Barbieri, C.E., Eedunuri, V.K., Chew, S.A., Zimmermann, M., Bond, R., Shou, J., Li, C., *et al.* (2013). Prostate cancer-associated mutations in speckle-type POZ protein (SPOP) regulate steroid receptor coactivator 3 protein turnover. *Proc Natl Acad Sci U S A* **110**, 6997-7002.
- Groner, A.C., Cato, L., de Tribolet-Hardy, J., Bernasocchi, T., Janouskova, H., Melchers, D., Houtman, R., Cato, A.C.B., Tschopp, P., Gu, L., *et al.* (2016). TRIM24 Is an Oncogenic Transcriptional Activator in Prostate Cancer. *Cancer Cell* **29**, 846-858.
- Guerrero-Puigdevall, M., Fernandez-Fuentes, N., and Frigola, J. (2021). Stabilisation of half MCM ring by Cdt1 during DNA insertion. *Nat Commun* **12**, 1746.
- Kuipers, M.A., Stasevich, T.J., Sasaki, T., Wilson, K.A., Hazelwood, K.L., McNally, J.G., Davidson, M.W., and Gilbert, D.M. (2011). Highly stable loading of Mcm proteins onto chromatin in living cells requires replication to unload. *J Cell Biol* **192**, 29-41.
- Marzahn, M.R., Marada, S., Lee, J., Nourse, A., Kenrick, S., Zhao, H., Ben-Nissan, G., Kolaitis, R.M., Peters, J.L., Pounds, S., *et al.* (2016). Higher-order oligomerization promotes localization of SPOP to liquid nuclear speckles. *EMBO J* **35**, 1254-1275.
- Matson, J.P., Dumitru, R., Coryell, P., Baxley, R.M., Chen, W., Twaroski, K., Webber, B.R., Tolar, J., Bielinsky, A.K., Purvis, J.E., *et al.* (2017). Rapid DNA replication origin licensing protects stem cell pluripotency. *Elife* **6**.
- McGarry, T.J., and Kirschner, M.W. (1998). Geminin, an inhibitor of DNA replication, is degraded during mitosis. *Cell* **93**, 1043-1053.
- Mukherjee, P., Cao, T.V., Winter, S.L., and Alexandrow, M.G. (2009). Mammalian MCM loading in late-G(1) coincides with Rb hyperphosphorylation and the transition to post-transcriptional control of progression into S-phase. *PLoS One* **4**, e5462.

Nguyen, V.Q., Co, C., and Li, J.J. (2001). Cyclin-dependent kinases prevent DNA re-replication through multiple mechanisms. *Nature* *411*, 1068-1073.

Tanaka, T., Knapp, D., and Nasmyth, K. (1997). Loading of an Mcm protein onto DNA replication origins is regulated by Cdc6p and CDKs. *Cell* *90*, 649-660.

Ticau, S., Friedman, L.J., Ivica, N.A., Gelles, J., and Bell, S.P. (2015). Single-molecule studies of origin licensing reveal mechanisms ensuring bidirectional helicase loading. *Cell* *161*, 513-525.

Wang, Z., Song, Y., Ye, M., Dai, X., Zhu, X., and Wei, W. (2020). The diverse roles of SPOP in prostate cancer and kidney cancer. *Nat Rev Urol* *17*, 339-350.

Yardimci, H., and Walter, J.C. (2014). Prereplication-complex formation: a molecular double take? *Nat Struct Mol Biol* *21*, 20-25.

Yin, W.C., Satkunendran, T., Mo, R., Morrissy, S., Zhang, X., Huang, E.S., Uuskula-Reimand, L., Hou, H., Son, J.E., Liu, W., *et al.* (2019). Dual Regulatory Functions of SUFU and Targetome of GLI2 in SHH Subgroup Medulloblastoma. *Dev Cell* *48*, 167-183 e165.

Yuan, Z., Schneider, S., Dodd, T., Riera, A., Bai, L., Yan, C., Magdalou, I., Ivanov, I., Stillman, B., Li, H., *et al.* (2020). Structural mechanism of helicase loading onto replication origin DNA by ORC-Cdc6. *Proc Natl Acad Sci U S A* *117*, 17747-17756.

Zhang, P., Wang, D., Zhao, Y., Ren, S., Gao, K., Ye, Z., Wang, S., Pan, C.W., Zhu, Y., Yan, Y., *et al.* (2017). Intrinsic BET inhibitor resistance in SPOP-mutated prostate cancer is mediated by BET protein stabilization and AKT-mTORC1 activation. *Nat Med* *23*, 1055-1062.

Zhuang, M., Calabrese, M.F., Liu, J., Waddell, M.B., Nourse, A., Hammel, M., Miller, D.J., Walden, H., Duda, D.M., Seyedin, S.N., *et al.* (2009). Structures of SPOP-substrate complexes: insights into molecular architectures of BTB-Cul3 ubiquitin ligases. *Mol Cell* *36*, 39-50.

REVIEWER COMMENTS

Reviewer #1 (Remarks to the Author):

The manuscript "SPOP mutation induces replication over-firing by impairing Geminin ubiquitylation and triggers catastrophe upon ATR inhibition" is an interesting, thorough and comprehensive story about SPOP, Geminin and origin firing.

I would like to thank the authors for thorough response to my comments.

The manuscript in its previous version was already really nice but I believe that the changes introduced upon reviewers comments improved it. I am happy to recommend it for publication without any additional changes.

Reviewer #3 (Remarks to the Author):

The authors have addressed my comments and they appropriately corrected their mention of the OCCM loading intermediate. The authors decided to use a different structure from what was done in their first submission for their atomic modelling. They used the structure of a pre-insertion loading intermediate of OCCM, obtained by truncating the C-terminal WD domain of Mcm6 (6WGG), which is the latest ORC-Cdc6-Cdt1-MCM complex published. Mentioning that this structural intermediate indeed precedes stable OCCM formation during the helicase loading process seems appropriate. In other words, the structure used should be described correctly.

Apart from this small point, I think this work should now be published in Nature Communications.

Authors' Response to Reviewers' Comments on Manuscript NCOMMS-21-03998R1

We very much thank the reviewers for the positivity and insightful comments for the improvement of our manuscript.

Reviewer #1 (Remarks to the Author):

The manuscript "SPOP mutation induces replication over-firing by impairing Geminin ubiquitylation and triggers catastrophe upon ATR inhibition" is an interesting, thorough and comprehensive story about SPOP, Geminin and origin firing.

I would like to thank the authors for thorough response to my comments.

The manuscript in its previous version was already really nice but I believe that the changes introduced upon reviewers comments improved it. I am happy to recommend it for publication without any additional changes.

Response: We thank the Reviewer for the enthusiasm and recommendation for the publication of our manuscript without any additional changes.

Reviewer #3 (Remarks to the Author):

The authors have addressed my comments and they appropriately corrected their mention of the OCCM loading intermediate. The authors decided to use a different structure from what was done in their first submission for their atomic modelling. They used the structure of a pre-insertion loading intermediate of OCCM, obtained by truncating the C-terminal WD domain of Mcm6 (6WGG), which is the latest ORC-Cdc6-Cdt1-MCM complex published. Mentioning that this structural intermediate indeed precedes stable OCCM formation during the helicase loading process seems appropriate. In other words, the structure used should be described correctly.

Apart from this small point, I think this work should now be published in Nature Communications.

Response: We are glad that the Reviewer agree that we have addressed his/her comments. We also thank the Reviewer for pointing out that the OCCM structure used should be described correctly. We have corrected the description as indicated below:

“Current structural knowledge of the replication origin and origin recognition complex formation is mostly derived from studies in budding yeast (Yuan et al., Proc Natl Acad Sci U S A, 2020; Zhai et al., Nat Struct Mol Biol, 2017). A single-particle cryo-EM structure of a pre-insertion loading intermediate of ORC-Cdc6-Cdt1-MCM (OCCM) complex from yeast was recently obtained by truncating the C-terminal WD domain of Mcm6 (PDB 6WGG) (Yuan et al., Proc Natl Acad Sci U S A, 2020). This structural intermediate, which precedes OCCM formation during the helicase loading process, shows how Cdt1 associates with the MCM complex. Despite low sequence similarities, Cdt1 from human (hCdt1) and yeast (yCdt1) both have conserved winged-helix domains (WHD) in their middle (M-WHD) and C-terminal (C-WHD) regions (Supplementary Fig. 3f) (Pozo and Cook, Genes (Basel), 2016). Because of these structural

similarities, we built a model by using the crystal structure of the Geminin-bound M-WHD region of hCdt1 (PDB 2WVR) (De Marco et al., Proc Natl Acad Sci U S A, 2009) to replace the corresponding yCdt1 region in the aforementioned OCCM intermediate complex (PDB 6WGG) (Yuan et al., Proc Natl Acad Sci U S A, 2020). The hCdt1 and yCdt1 M-WHD structures can be overlaid without any steric interference on the MCM complex (Supplementary Fig. 3g).”

References

- De Marco, V., Gillespie, P.J., Li, A., Karantzelis, N., Christodoulou, E., Klompaker, R., van Gerwen, S., Fish, A., Petoukhov, M.V., Iliou, M.S., *et al.* (2009). Quaternary structure of the human Cdt1-Geminin complex regulates DNA replication licensing. *Proc Natl Acad Sci U S A* *106*, 19807-19812.
- Pozo, P.N., and Cook, J.G. (2016). Regulation and Function of Cdt1; A Key Factor in Cell Proliferation and Genome Stability. *Genes (Basel)* *8*.
- Yuan, Z., Schneider, S., Dodd, T., Riera, A., Bai, L., Yan, C., Magdalou, I., Ivanov, I., Stillman, B., Li, H., *et al.* (2020). Structural mechanism of helicase loading onto replication origin DNA by ORC-Cdc6. *Proc Natl Acad Sci U S A* *117*, 17747-17756.
- Zhai, Y., Cheng, E., Wu, H., Li, N., Yung, P.Y., Gao, N., and Tye, B.K. (2017). Open-ringed structure of the Cdt1-Mcm2-7 complex as a precursor of the MCM double hexamer. *Nat Struct Mol Biol* *24*, 300-308.